# ANOCE: ANALYSIS OF CAUSAL EFFECTS WITH MULTIPLE MEDIATORS VIA CONSTRAINED STRUCTURAL LEARNING

**Hengrui Cai**
North Carolina State University
Raleigh, USA
`hcai5@ncsu.edu`

**Rui Song**
North Carolina State University
Raleigh, USA
`rsong@ncsu.edu`

**Wenbin Lu**
North Carolina State University
Raleigh, USA
`wlu4@ncsu.edu`

## ABSTRACT

In the era of causal revolution, identifying the causal effect of an exposure on the outcome of interest is an important problem in many areas, such as epidemics, medicine, genetics, and economics. Under a general causal graph, the exposure may have a direct effect on the outcome and also an indirect effect regulated by a set of mediators. An analysis of causal effects that interprets the causal mechanism contributed through mediators is hence challenging but on demand. To the best of our knowledge, there are no feasible algorithms that give an exact decomposition of the indirect effect on the level of individual mediators, due to common interaction among mediators in the complex graph. In this paper, we establish a new statistical framework to comprehensively characterize causal effects with multiple mediators, namely, ANalysis Of Causal Effects (ANOCE), with a newly introduced definition of the mediator effect, under the linear structure equation model. We further propose a constrained causal structure learning method by incorporating a novel identification constraint that specifies the temporal causal relationship of variables. The proposed algorithm is applied to investigate the causal effects of 2020 Hubei lockdowns on reducing the spread of the coronavirus in Chinese major cities out of Hubei.

## 1 INTRODUCTION

In the era of causal revolution, identifying the causal effect of an exposure on the outcome of interest is an important problem in many areas, such as epidemics (Hernán, 2004), medicine (Hernán et al., 2000), education (Card, 1999), and economics (Panizza & Presbitero, 2014). Under a general causal graph, the exposure may have a direct effect on the outcome and also an indirect effect regulated by a set of mediators (or intermediate variables). For instance, during the outbreak of Coronavirus disease 2019 (COVID-19), the Chinese government has taken extreme measures to stop the virus spreading such as locking Wuhan down on Jan 23rd, 2020, followed by 12 other cities in Hubei, known as the "2020 Hubei lockdowns". This approach (viewed as the exposure), directly blocked infected people leaving from Hubei; and also stimulated various quarantine measures taken by cities outside of Hubei (as the mediators), which further decreased the migration countrywide in China, and thus indirectly control the spread of COVID-19. Quantifying the causal effects of 2020 Hubei lockdowns on reducing the COVID-19 spread regulated by different cities outside Hubei is challenging but of great interest for the current COVID-19 crisis. An analysis of causal effects that interprets the causal mechanism contributed via individual mediators is thus very important.

Many recent efforts have been made on studying causal effects that are regulated by mediators. Chakrabortty et al. (2018) specified the individual mediation effect in a sparse high-dimensional causal graphical model. However, the sum of marginal individual mediation effect is not equal to the effect of all mediators considered jointly (i.e. the indirect effect) due to the common interaction among mediators (VanderWeele & Vansteelandt, 2014). Here, 'interaction' means that there exists at

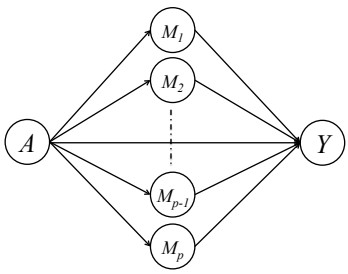
(a) A causal graph with parallel mediators.

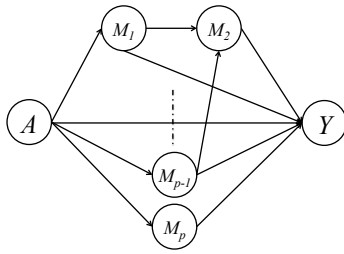
(b) A causal graph with interacted mediators.

Figure 1: Illustration of causal graphs with different types of mediators, where $A$ is the exposure, $\{M_1, \cdots, M_p\}$ are mediators, and $Y$ is the outcome of interest.

least one mediator that is regulated by other mediator(s) (see Figure 1b for illustration), in contrast to the simple 'parallel' case (shown in Figure 1a). Vansteelandt & Daniel (2017) considered an exact decomposition of the indirect effect with a two-mediator setting based on the conditional densities of mediators, while there was no feasible algorithm provided to solve their proposed expressions yet. Therefore, a new framework with a computational friendly algorithm that gives an exact decomposition of the indirect effect on the level of individual mediators is desired under the complex causal network.

To estimate the underlying causal network, structure learning algorithms of the directed acyclic graph (DAG) are widely used. Popular methods such as the PC algorithm (Spirtes et al., 2000) that uses conditional independence tests to examine the existence of edges between each pair of variables, require strong assumptions and thus have no guarantee in the finite sample regime. Recently, Zheng et al. (2018) opened up another class of causal discovery methods by directly formulating a pure optimization problem over real metrics with a novel characteristic of the acyclicity. Yu et al. (2019) further extended Zheng et al. (2018)'s work with a deep generative model, and showed better performance on the structure learning with weaker assumptions on the noise. See more follow-up works in Lachapelle et al. (2019) and Zhu & Chen (2019). However, the current cutting-edge methods neglect the temporal causal relationship among variables, and thus cannot appropriately represent the causal network with pre-specified exposure and outcome.

In this paper, we consider establishing a new statistical framework to comprehensively characterize causal effects with multiple mediators, namely, ANalysis Of Causal Effects (ANOCE), under the linear structure equation model (LSEM). Specifically, we propose two causal effects on the level of individual mediators, the natural direct effect and the natural indirect effect for a mediator, denoted as $DM$ and $IM$, respectively. Our proposed $DM$ can be interpreted as the direct effect of a particular mediator on the outcome that is not regulated by other mediators, while the $IM$ is the indirect effect of the mediator controlled by its descendant mediators. We prove that the $DM$ is valid in the sense that it exactly decomposes the indirect effect of the exposure on the outcome, followed by an ANOCE table to explain different sources of causal effects. To bridge the cutting-edge graphical learning approaches with the temporal causal relationship of variables, we extend the variational auto-encoder (VAE) framework in Yu et al. (2019) with a novel identification constraint that specifies the topological order of the exposure and the outcome. The proposed constrained VAE algorithm is then used to estimate causal effects defined in our ANOCE table, named as 'ANOCE-CVAE'.

Our contributions can be summarized in the following three aspects:
• 1). Conceptually, we define different sources of causal effects through mediators with a newly introduced definition of direct and indirect mediator effects, and give an exact decomposition of the indirect effect on the level of individual mediators, under the linear structure equation model.
• 2). Methodologically, we incorporate the background knowledge (the temporal causal relationship among variables) when using an optimization approach to the causal discovery. Such prior knowledge can be generalized for any measured variable and on the possible set of their parents. Our proposed constrained structural learning can be easily extended to other score-based algorithms.
• 3). Practically, extensive simulations are conducted to demonstrate the empirical validity of the

proposed algorithm and its competitive performance among existing causal discovery algorithms. Our method is applied to investigate the causal effects of 2020 Hubei lockdowns on reducing the COVID-19 spread in China, by quantifying the individual effect for each city.

## 2 BACKGROUND AND RELATED WORK

### 2.1 GRAPH TERMINOLOGY

Consider a graph $\mathcal{G} = (X, E)$ with a node set $X$ and an edge set $E$. There is at most one edge between any pair of nodes. If there is an edge between $X_i$ and $X_j$, then $X_i$ and $X_j$ are adjacent. A node $X_i$ is said to be a parent of $X_j$ if there is a directed edge from $X_i$ to $X_j$. Let the set of all parents of node $X_j$ in $\mathcal{G}$ as $PA_{X_j}(\mathcal{G})$. A path from $X_i$ to $X_j$ in $\mathcal{G}$ is a sequence of distinct vertices, $\pi \equiv \{a_0, a_1, \cdots, a_L\} \subset V$ such that $a_0 = X_i$, and $a_L = X_j$. A directed path from $X_i$ to $X_j$ is a path between $X_i$ and $X_j$ where all edges are directed toward $X_j$. A directed cycle is formed by the directed path from $X_i$ to $X_j$ together with the directed edge $X_j$ to $X_i$. A directed graph that does not contain directed cycles is called a directed acyclic graph (DAG). A directed graph is acyclic if and only if it has a topological ordering. Suppose a DAG $\mathcal{G} = (X, E)$ that characterizes the causal relationship among $|X| = d$ nodes, where $X = [X_1, X_2, \cdots, X_d]^\top$ represents a random vector and an edge $X_i \to X_j$ means that $X_i$ is a direct cause of $X_j$. Let $B = \{b_{i,j}\}_{1 \le i \le d, 1 \le j \le d}$ be a $d \times d$ matrix, where $b_{i,j}$ is the weight of the edge $X_i \to X_j \in E$, and $b_{i,j} = 0$ otherwise. Then, we say that $\mathcal{G} = (X, B)$ is a weighted DAG with the node set $X$ and the weighted adjacency matrix $B$ (the edge set $E$ is nested in $B$).

### 2.2 RELATED WORK

Our work connects to the literature of the causal graphical model. Pearl et al. (2009) provided a comprehensive review of recent advances in the analysis of causes and counterfactuals using 'do-operator' by graphical methods. Later, Maathuis et al. (2009) started to use an unknown DAG without hidden variables to estimate the causal effects from the high-dimensional observational data. Nandy et al. (2017) extended the work of Maathuis et al. (2009) with the linear structure equation model, followed by the individual mediation effect defined in Chakrabortty et al. (2018). All of these models rely on the PC algorithm to search the Markov equivalence class of the partial DAG, and usually require strong assumptions due to the computational limit. Our ANOCE is established under the same causal structure of Chakrabortty et al. (2018) but without sparsity and normality assumptions.

Wide literature on causal discovery can be summarized in three classes. The first type focuses on local conditional independence tests to find a causal skeleton and then determine the orientation of edges, such as the well-known PC algorithm (Spirtes et al., 2000; Kalisch & Bühlmann, 2007). However, testing the conditional independence of continuous variables is not easy (Shah & Peters, 2018). The second class specifies properly functional causal models with additional assumptions on data distribution, including the ICA-LiNGAM (Shimizu et al., 2006) and the causal additive model (CAM) (Bühlmann et al., 2014). The last class, the score-based method, includes the greedy equivalence search (GES) (Chickering, 2002) and the fast GES (fGES) (Ramsey et al., 2017) that use for example Bayesian scores in searching a space of causal models. Recently, Zheng et al. (2018) opened up another track of score-based methods by constructing an optimization with an acyclicity constraint under the LSEM, i.e. the NOTEARS. A follow-up work using a VAE parameterized by a graph neural network that generalizes LSEM was proposed in Yu et al. (2019) with a more computational friendly constraint, namely DAG-GNN. Also see Zhu & Chen (2019) and Lachapelle et al. (2019) for other cutting-edge structural learning methods.

The improvement of our ANOCE-CVAE over the state-of-the-arts is as follows. We consider a new constrained structural learning, by incorporating the background knowledge (the temporal causal relationship among variables) into the score-based algorithms. We formulated such prior information as the identification constraint and add it as the penalty term in the objective function for the causal discovery. In this paper, we typically extend the DAG-GNN for an illustration. Note that the proposed constraint is not limited to the DAG-GNN and can be easily extended to other score-based algorithms.

## 3 ANALYSIS OF CAUSAL EFFECTS

### 3.1 STATISTICAL FRAMEWORK AND ASSUMPTIONS

Let $A$ be the exposure/treatment, $M = [M_1, M_2, \cdots, M_p]^\top$ be mediators with dimension $p$, and $Y$ be the outcome of interest. Suppose there exists a weighted DAG $\mathcal{G} = (X, B)$ that characterizes the causal relationship among $X = [A, M^\top, Y]^\top$, where the dimension of $X$ is $d = p + 2$. Let $Y^*(A = a, M = m)$ be the potential outcome that would be observed after receiving treatment $a$ and setting mediators as $m$, and $M^*(A = a)$ be the potential mediators that would be observed after receiving treatment $a$. As standard in the causal inference (Rosenbaum & Rubin, 1983), we assume that there is no unmeasured confounder: (A1) the effect of the treatment $A$ on the outcome $Y$ is unconfounded, i.e. $Y^*(A = a, M = m) \perp A, \forall a, m$; (A2) the effect of the treatment $A$ on the mediators $M$ is unconfounded, i.e. $M^*(A = a) \perp A, \forall a$; (A3) the effect of the mediators $M$ on the outcome $Y$ is unconfounded given the treatment $A$, i.e. $Y^*(A = a, M = m) \perp M | A, \forall a, m$. In addition, as standard in the graphical causal discovery, we also make the Markov condition, the faithfulness condition, causal sufficiency assumption, and the linear structural equation model (LSEM) such that $X = [A, M^\top, Y]^\top$ characterized by the pair $(\mathcal{G}, \epsilon)$ is generated by

$$X = B^\top X + \epsilon, \tag{1}$$

where $\epsilon$ is a random vector of jointly independent error variables.

Denote all directed paths in $\mathcal{G}$ that start with the exposure $A$ and end with the outcome $Y$ as set $\{\pi_{AY}(\mathcal{G})\}$. If there exists at least one directed path $\pi^* \in \{\pi_{AY}(\mathcal{G})\}$ such that the length of $\pi^*$ is larger than 2, we say there is an interaction among mediators, as shown in Figure 1b; otherwise, we call mediators are 'parallel' as shown in Figure 1a. In this paper, we consider all possible causal structures with multiple mediators under assumptions (A1-A3).

We next give the total effect ($TE$), the natural direct effect that is not mediated by mediators ($DE$), and the natural indirect effect that is regulated by mediators ($IE$) defined in Pearl et al. (2009).

**Definition 3.1** *(Pearl et al., 2009)*

$$TE = \partial E\{Y|do(A = a)\}/\partial a = E\{Y|do(A = a + 1)\} - E\{Y|do(A = a)\},$$
$$DE = E\{Y|do(A = a + 1, M = m^{(a)})\} - E\{Y|do(A = a)\},$$
$$IE = E\{Y|do(A = a, M = m^{(a+1)})\} - E\{Y|do(A = a)\},$$

*where $do(A = a)$ is a mathematical operator to simulate physical interventions that hold $A$ constant as $a$ while keeping the rest of the model unchanged, which corresponds to remove edges into $A$ and replace $A$ by the constant $a$ in $\mathcal{G}$. Here, $m^{(a)}$ is the value of $M$ if setting $do(A = a)$, and $m^{(a+1)}$ is the value of $M$ if setting $do(A = a + 1)$. Refer to Pearl et al. (2009) for more details of 'do-operator'.*

Note that in the assumed linear model, the slope of the line is the same everywhere; for convenience and simplicity, we use $a$ and $a + 1$ to present the change of the treatment of 1 in the definition.

### 3.2 NATURAL DIRECT AND INDIRECT EFFECT FOR INDIVIDUAL MEDIATORS

We first give the definition of the natural direct effect for an individual mediator ($DM$).

**Definition 3.2** *Natural direct effect for $M_i$:*

$$DM_i = \Big[E\{M_i|do(A = a + 1)\} - E\{M_i|do(A = a)\}\Big] \\ \times \Big[E\{Y|do(A = a, M_i = m_i^{(a)} + 1, \Omega_i = o_i^{(a)})\} - E\{Y|do(A = a)\}\Big], \tag{2}$$

*where $m_i^{(a)}$ is the value of $M_i$ when setting $do(A = a)$, $\Omega_i = M \setminus M_i$ is the set of mediators except $M_i$, and $o_i^{(a)}$ is the value of $\Omega_i$ when setting $do(A = a)$.*

**Remark 3.1** *From Definition 3.2, the natural direct effect for $M_i$ is the product of the total effect of the treatment $A$ on the mediator $M_i$ and the direct effect of the mediator $M_i$ on the outcome $Y$. The*

*second multiplier is in line with the classical meaning of 'natural' in the causal inference literature (Pearl et al., 2009). Thus, the DM can be interpreted as the causal effect through a particular mediator from the treatment on the outcome that is not regulated by its descendent mediators.*

The natural indirect effect for an individual mediator ($IM$) can be defined similarly.

**Definition 3.3** *Natural indirect effect for $M_i$:*

$$IM_i = \Big[E\{M_i|do(A = a + 1)\} - E\{M_i|do(A = a)\}\Big]$$
$$\times \Big[E\{Y|do(A = a, M_i = m_i^{(a)} + 1)\} - E\{Y|do(A = a, M_i = m_i^{(a)} + 1, \Omega_i = o_i^{(a)})\}\Big].$$

**Remark 3.2** *The second multiplier in the $IM_i$ captures the indirect effect of a particular mediator on the outcome regulated by its descendent mediators. We show the individual mediation effect ($\eta$) in Chakrabortty et al. (2018) can be decomposed into the $DM$ and the $IM$ in Section F in the appendix when the LSEM assumption holds, i.e., $\eta_i = DM_i + IM_i$, for $i$-th mediator.*

Next, we give explicit expressions of defined causal effects under the LSEM. Specifically, we can write the linear structural model 1 under assumptions (A1-A3) as

$$\begin{bmatrix} A \\ M \\ Y \end{bmatrix} = B^\top \begin{bmatrix} A \\ M \\ Y \end{bmatrix} + \epsilon = \begin{bmatrix} 0 & \mathbf{0}_{p\times 1} & 0 \\ \boldsymbol{\alpha} & B_M^\top & 0 \\ \gamma & \boldsymbol{\beta}^\top & 0 \end{bmatrix} \begin{bmatrix} A \\ M \\ Y \end{bmatrix} + \begin{bmatrix} \epsilon_A \\ \epsilon_{M_p} \\ \epsilon_Y \end{bmatrix}, \tag{3}$$

where $\gamma$ is a scalar, $\boldsymbol{\alpha}$, $\boldsymbol{\beta}$, and $\mathbf{0}_{p\times 1}$ are $p \times 1$ vectors, $B_M$ is a $p \times p$ matrix, and $\epsilon \equiv [\epsilon_A, \epsilon_M^\top, \epsilon_Y]^\top$. Here, $\gamma$ presents the weight of the edge $A \to Y$, the $i$-th element of $\boldsymbol{\alpha}$ corresponds to the weight of the edge $A \to M_i$, and the $i$-th element of $\boldsymbol{\beta}$ is the weight of the edge $M_i \to Y$. Note that by assumptions (A1-A3), we have the exposure $A$ has no parents and the outcome $Y$ has no descendants, so equivalently, the first row and the last column of $B^\top$ are all zeros (i.e., the first column and the last row of $B$ are all zeros). Notice the exposure can be presented by its own noise, i.e., $A = \epsilon_A$, since $A$ has no parents, so any exposure (with arbitrary noise distribution) will satisfy the LSEM assumption.

Next, we obtain expressions of causal effects under the LSEM in the following theorem. The proof can be found in Section G.1 of the appendix.

**Theorem 3.1** *Under assumptions (A1-A3) and Model 1, we have:*
*1). the natural direct effect is $DE = \gamma$;*
*2). the natural indirect effect is $IE = \boldsymbol{\beta}^\top (I_p - B_M^\top)^{-1} \boldsymbol{\alpha}$, where $I_p$ is a $p \times p$ identity matrix;*
*3). the total effect of A on Y is $TE = \gamma + \boldsymbol{\beta}^\top (I_p - B_M^\top)^{-1} \boldsymbol{\alpha}$;*
*4). the natural direct effect of $M_i$ on Y is $DM_i = \boldsymbol{\beta}_i\{(I_p - B_M^\top)^{-1}\boldsymbol{\alpha}\}_i$, where $\boldsymbol{\beta}_i$ is the $i$-th element of $\boldsymbol{\beta}$ corresponding to the weight of $M_i \to Y$, and $\{(I_p - B_M^\top)^{-1}\boldsymbol{\alpha}\}_i$ is the $i$-th element of $(I_p - B_M^\top)^{-1}\boldsymbol{\alpha}$ as the total effect of A on $M_i$, i.e. $E\{M_i|do(A = a + 1)\} - E\{M_i|do(A = a)\}$.*

**Remark 3.3** *One may refer to section A in the appendix for the invertibility of $I_p - B_M^\top$. Also, a toy example is provided in section E to illustrate how to manually compute the causal effects defined above. Note that there is no explicit expression of the $IM$ due to the complex interaction among mediators, while we provide its theoretical form in Section G.2 with its numerical form in Section B.*

Based on the result 2) and 4) in Theorem 3.1, the $IE$ can be presented as an additive form of $DM$s, as shown in Theorem 3.2. Thus, the proposed natural direct effect of individual mediators is valid in the sense that it exactly decomposes the indirect effect of the exposure on the outcome.

**Theorem 3.2** *Under assumptions (A1-A3) and Model 1, the $IE$ can be decomposed through $DM$s:*

$$IE = \sum_{i=1}^{p} DM_i.$$

## 3.3 ANALYSIS OF CAUSAL EFFECTS TABLE

Based on the result $TE = DE + IE$ in Pearl et al. (2009) and Theorem 3.2, we summarize the defined causal effects and their relationship in Table 1 for the analysis of causal effects (ANOCE).

Firstly, the causal effect of $A$ on $Y$ has two sources, the direct effect from $A$ and the indirect effect via $p$ mediators $M$ $(M_1, \cdots, M_p)$. Next, the direct source has the degree of freedom $(d.f.)$ as 1, while the indirect source has $d.f.$ as $p$ from $p$ mediators. Note the true $d.f.$ of the indirect effect may be smaller than $p$, since $A$ may not be regulated by all mediators. Then, the causal effect for the direct source is the $DE$ and for the indirect source is the $IE$, where the $IE$ can be further decomposed into $p$ $DM$s and each component corresponds to the natural direct effect for a specific mediator. The last row in the table shows that the $DE$ and the $IE$ compose the total effect $TE$ with $d.f.$ as $p + 1$.

Table 1: Table of Analysis of Causal Effects (ANOCE Table).

| Source | | Degree of freedom | | Causal effects | |
|---|---|---|---|---|---|
| Direct effect from $A$ | | 1 | | $DE$ | |
| Indirect effect via $M$ | | $p$ | | $IE$ | |
| | $M_1$ | | 1 | | $DM_1$ |
| | $M_2$ | | 1 | | $DM_2$ |
| | $\vdots$ | | $\vdots$ | | $\vdots$ |
| | $M_p$ | | 1 | | $DM_p$ |
| Total | | $1 + p$ | | $TE$ | |

## 4 CONSTRAINED STRUCTURAL LEARNING FOR ANOCE

We next estimate the weighted adjacency matrix $B$ with our causal framework under the LSEM to calculate causal effects. To better capture the sampling distribution faithful to the DAG, we consider a deep generative model that generalizes the LSEM instead of using a regression that heavily relies on assumptions of noise (see more discussion in Section 2.2). Specifically, the LSEM 1 can be rewritten as $(I_{p+2} - B^\top)X = \epsilon$, where $I_{p+2}$ is a $(p+2) \times (p+2)$ identity matrix. Inversely, we have $X = (I_{p+2} - B^\top)^{-1}\epsilon$. Following the VAE architecture in Yu et al. (2019), we treat the random error $\epsilon$ as the independent latent variables to generate $X$, by two multilayer perceptrons as the encoder and the decoder, with weights denoted as $\theta$. We adopt their acyclicity constraint on $B$ as,

$$h_1(B) \equiv \mathrm{tr}\big[(I_{p+2} + tB \bullet B)^{p+2}\big] - (p+2) = 0, \tag{4}$$

where $\mathrm{tr}(\cdot)$ is the trace of a matrix, $t$ is a hyperparameter that depends on an estimation of the largest eigenvalue of $B$, and $\bullet$ denotes for the element-wise square.

Next, to incorporate the background knowledge of the temporal causal relationship among variables, we propose an identification constraint that indicates the topological order of the exposure and the outcome. As mentioned in Equation 3, under assumptions (A1-A3), the exposure $A$ has no parents, i.e. $PA_A(\mathcal{G}) = \varnothing$, and the outcome $Y$ has no descendants, i.e. $Y \notin PA_X(\mathcal{G})$. Or equivalently, we have the first column and the last row of $B$ should equal to zero. Therefore, the matrix $B$ must satisfy

$$h_2(B) \equiv \sum_{i=1}^{p+2} |b_{i,1}| + \sum_{j=2}^{p+2} |b_{p+2,j}| = 0, \tag{5}$$

where $b_{i,j}$ is the element of the matrix $B$ in $i$-th row and $j$-th column. The above constraint forces the topological order of the exposure as 1 while the outcome as $p + 2$, under which the DAG is searched within a restricted regime. The prior knowledge in 5 can be generalized for any measured variable and on the possible set of their parents, by connecting the topological order to the weighted matrix $B$.

Following Yu et al. (2019), the objective function is the evidence lower bound with two constraints:

$$\begin{cases} \min\limits_{B,\theta} \quad f(B,\theta) = \frac{1}{p+2} \sum_{i=1}^{p+2} D_{KL}\{q(\epsilon|X_i)||p(\epsilon)\} - E_{q(\epsilon|X_i)}\{\log p(X_i|\epsilon)\}, \\ s.t. \quad h_1(B) = 0 \quad \text{and} \quad h_2(B) = 0, \end{cases} \tag{6}$$

where $D_{KL}(\cdot||\cdot)$ is the Kullback-Leibler divergence, $p(\epsilon)$ is the prior distribution of $\epsilon$, $q(\epsilon|X_i)$ is the reconstructed empirical posterior distribution of $\epsilon$, and $p(X_i|\epsilon)$ is the likelihood function. Then, we have the loss function based on the augmented Lagrangian as

$$L_{c,d}(B,\theta,\lambda_1,\lambda_2) = f(B,\theta) + \lambda_1 h_1(B) + \lambda_2 h_2(B) + c|h_1(B)|^2 + d|h_2(B)|^2, \tag{7}$$

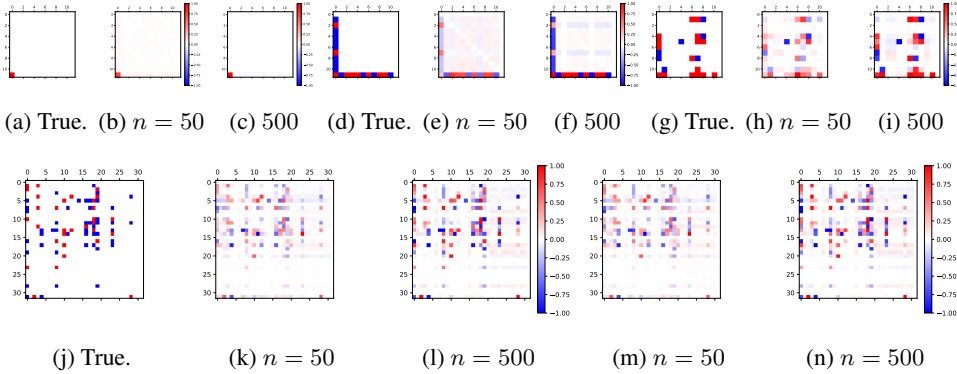

(a) True.  (b) $n = 50$  (c) 500  (d) True.  (e) $n = 50$  (f) 500  (g) True.  (h) $n = 50$  (i) 500

(j) True.  (k) $n = 50$  (l) $n = 500$  (m) $n = 50$  (n) $n = 500$

Figure 2: The averaged estimated $\widehat{B}^{\top}$ under the ANOCE-CVAE: (a-c) correspond to Scenario 1; (d-f) correspond to Scenario 2; (g-i) correspond to Scenario 3; (j) is the shared true DAG for Scenario 4 and $4^*$; (k-l) are results under Scenario 4; (m-n) are results under Scenario $4^*$.

where $\lambda_1$ and $\lambda_2$ are Lagrange multipliers, and $c$ and $d$ are penalty terms. To minimize the loss in 7 and satisfy both $h_1(B) = 0$ and $h_2(B) = 0$, we simultaneously update $\lambda_1$ and $\lambda_2$ and increase $c$ and $d$ to infinity, by modifying the basic technique in Yu et al. (2019). Here, the minimization can be solved using a blackbox stochastic optimization such as 'Adam' in Kingma & Ba (2014). Denote the estimated matrix as $\widehat{B}$ from the above constrained structural learning. Under Theorem 3.1, we can estimate causal effects in the ANOCE table based on the learned $\widehat{B}$. We name the above algorithm as ANOCE-CVAE, with a detail pseudocode provided in Section B.

**Remark 4.1** *We incorporate the temporal causal relationship among variables when using an optimization approach to the causal discovery. Such constrained structural learning is not limited to the VAE framework and can be extended to any score-based algorithms. For instance, one can add constraint 5 into the objective function in Zheng et al. (2018) or the reward in Zhu & Chen (2019).*

## 5 EXPERIMENTS

We conduct extensive simulation studies to investigate the proposed method on learning causal effects with multiple mediators, followed by a comparison to the popular structural learning algorithms. The dataset and the code are publicly available at `https://github.com/anoce-cvae/ANOCE-CVAE`.

### 5.1 SIMULATION STUDIES

Scenarios are generated as follows. In Scenario 1 to 3, we fix the dimension of $M$ as $p = 10$ while increasing the complexity of the true graph to examine the sensitivity of our algorithm to sparsity. Specifically, Scenario 1 is the simplest causal graph with only one edge ($A \rightarrow Y$) shown in Figure 2a; and Scenario 2 has a fully connected graph with independent mediators (corresponding to the parallel case, i.e. $B_M = \mathbf{0}_{p \times p}$) illustrated in Figure 2d. In Scenario 3, we consider interacted mediators such that $B_M \neq \mathbf{0}_{p \times p}$, as demonstrated in Figure 2g. For Scenario 4, we allow $p = 30$ with interacted mediators to examine the stability of our method under the high-dimensional setting. Here, the true DAGs in Scenarios 3 and 4 are generated from the Erdős-Reńyi (ER) model with an expected degree as 2 . Note that we consider fully identifiable models in the experiments so that it is meaningful to evaluate causal effects from the estimated graph. The synthetic datasets $\{A, M, Y\}$ are generated from Model 1 with Gaussian errors in Scenario 1-4. We also set $A \in \{-1, 1\}$ in Scenario 4 to show that our algorithm is capable to handle both discrete and continuous exposure, denoted as Scenario $4^*$. The sample size $n$ is chosen from $\{50, 500\}$ to be consistent with the scale of our real data. See more details of the data generation in Section C.1 and the implementation in Section C.2 in the appendix.

The averaged estimated matrix $\widehat{B}^{\top}$ over 100 replications under the proposed ANOCE-CVAE is illustrated in Figure 2. The numerical results are summarized in Table 2 (for Scenario 1 to 3) and

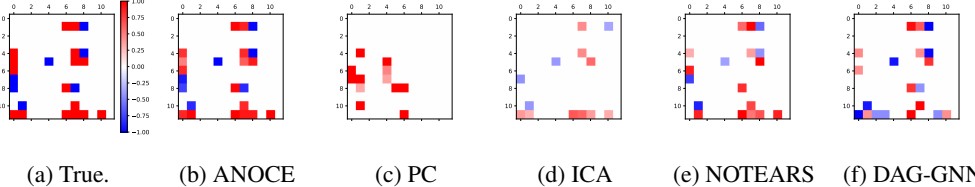

| (a) True. | (b) ANOCE | (c) PC | (d) ICA | (e) NOTEARS | (f) DAG-GNN |

Figure 3: The averaged estimated $\widehat{B}^\top$ for Scenario 3 (ER2) with the graph threshold as $0.3$.

Table 3 (for Scenario 4 and $4^*$) in the appendix, including the bias of the estimated $TE$, $DE$, $IE$, $DM$ and $IM$ for each mediator with their standard error. It can be observed that our proposed method could correctly identify most of the edges in the causal graph when $n = 500$ in almost all cases. Based on Table 2 and 3, the estimated causal effects are close to the true values as the sample size increases, indicating the good performance of our proposed method on identifying the causal effects regardless of the sparsity, the distribution of the exposure, and the dimension of mediators.

## 5.2 COMPARISON

We next compare our approach against the PC (Spirtes et al., 2000), the ICA-LiNGAM (Shimizu et al., 2006), the NOTEARS (Zheng et al., 2018), and the DAG-GNN (Yu et al., 2019). Random graphs are generated from both the ER and the Scale-Free (SF) networks with the expected degree as 1, 2, and 4, denoted as Cases ER1, ER2, ER4, SF1, SF2, and SF4, respectively. To be consistent with Section 5.1, we refer Scenario 3 (generated by the ER with the degree as 2) as ER2, and set $d = 12$ (i.e. $p = 10$) with Gaussian errors for other five cases, under $n = 500$. Details of the data generation and the implementation of each method are reported in Section C.1 and C.2. Here, we use a graph threshold as 0.3 (commonly used in other methods) and 0.4 to prune the noise edges for a fair comparison. The averaged estimated matrix $\widehat{B}^\top$ over 100 replications under different methods is shown in Figure 3 with a graph threshold as 0.3 for Scenario 3 (i.e., Case ER2) as an illustration. See other cases in Figures 5 to 15 in the appendix. All the numerical results of six cases are reported in Tables 4 and 5 in the appendix, including the false discovery rate, the true positive rate, and the structural Hamming distance. It is shown our algorithm performs the best among five methods in most cases, followed by the NOTEARS and the DAG-GNN. The comparison studies not only support the choice of the extension on the score-based algorithm (by comparing the results of the NOTEARS and the DAG-GNN with other methods), but also validate the improvement of our method over the DAG-GNN by introducing the background knowledge in the causal discovery.

## 6 REAL DATA ANALYSIS: COVID-19 OUTBREAK

From early Jan 2020 to late Feb 2020 (the Spring Festival period), COVID-19 spread to every province-level division of China, exacerbated by the Chinese new year migration and human to human transmission. The Chinese Government locked Hubei down on Jan 24th, which directly blocked infected people leaving from Hubei, and also indirectly control the spread of COVID-19. Here, for simplicity, we attribute the causal effects of the measures taken by cities outside of Hubei to the original and main action of interest, i.e. Hubei lockdowns. Thus, the COVID-19 example satisfies the considered causality framework for studying the causal mediator effects.

We collect the data from the National Health Commission (NHC) of China and Baidu Qianxi for analysis. Specifically, let the exposure $A$ as if Hubei is on lockdown, 0 for unlocked (before and on Jan 23rd), and 1 for locked (on and after Jan 24th). We select 30 candidate cities outside Hubei that contain most potential infected people, as mediators $M$. The daily migration scale index (MSI) of each city is used as the value of each mediator, which is the migration magnitude of large groups of people from one geographical area to another (Chen et al., 2020) and is comparable among cities. Lastly, we use the daily increase rate of confirmed cases out of Hubei to characterize the severity of the virus spreading with a one-week delay (due to the diagnose and incubation period of COVID-19 (Lauer et al., 2020)): $Y_t = \frac{\text{Confirmed cases out of Hubei}_{t+8} - \text{Confirmed cases out of Hubei}_{t+7}}{\text{Confirmed cases out of Hubei}_{t+7}}$. Here, the time $t$ starts from Jan 12th to Feb 20th, 2020, since Jan 19th, 2020 is the earliest date with an available number of

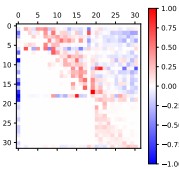

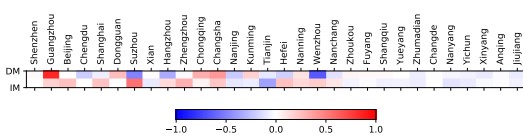

(a) The estimated weighted matrix for $B^\top$.

(b) The estimated (in)direct effects for selected cities.

Figure 4: The analysis of the causal effects of 2020 Hubei lockdowns on reducing the COVID-19 spread in China regulated by Chinese major cities outside Hubei.

confirmed cases out of Hubei (to compute $Y_{t=1}$ on Jan 12th), and after Feb 20th, 2020, the pandemic was under control outside Hubei with the evidence of the work resumption in China. The final dataset yields a total of 38 records. More details of data collection can be found in Section D of the appendix.

The proposed algorithm is applied to the COVID-19 data with 100 replications by setting different random seeds in the neural network. The estimated weighted adjacency matrix is shown in Figure 4a, with the detailed ANOCE table reported in Table 6 in the appendix. The total effect of 2020 Hubei lockdowns on the daily increase rate of confirmed cases outside Hubei as -0.497, where the direct effect is -0.078 and the indirect effect is -0.419. In other words, by locking Hubei down, China successfully reduced 49.7% of the daily new cases outside Hubei; 84% of which is the indirect effect contributed via the reduced migration of cities (the mediators) out of Hubei, and the rest 16% owes to the direct effect of Hubei lockdowns since infected people were constrained in Hubei after the lockdown. Thus, the lockdown is effective in reducing the COVID-19 spread in China.

The total indirect effect of the lockdown can be further broken down by cities' direct effects ($DM$s, corresponds to the intensity of transmission within a particular city). We compare cities' $DM$s with their associated indirect effects ($IM$s, describes the secondary migration from a particular city to other places) in Figure 4b, where a positive effect means spreading the virus while negative means control. Note that the selected 30 cities are ordered by their cumulative MSI during the data period. ● 1). From Figure 4b, the majority of cities have a negative $DM$ (colored in blue), which implies the infection within cities outside Hubei have been effectively controlled under the lockdown. ● 2). There are more cities with a positive $IM$s (red), which is in line with the intuition that the secondary migration among cities may exacerbate the pandemic. ● 3). The positive effects (red) are more likely located at the first 20 nodes, which corresponds to the cities with large MSI, while the last 10 cities with relatively small MSI are almost all blue. This accords with the migration peak among big cities during the Spring Festival period that aggravated the spread of the virus.

## 7 CONCLUSION

We conclude our paper with the following discussions. First, the proposed $DM$ can be extended beyond the LSEM assumption. A generalized definition of the $DM$ from a graphical perspective is given in Section F.2 of the appendix without the LSEM. Second, due to possibly unmeasured confounders in our real data, such as cities' features and periodic effect, we may consider extending our model with a new topological order that contains confounders for a wider utility, such as forcing the topological order of $k$ confounders as 1 to $k$ followed by the exposure as $1 + k$. Third, our proposed identification constraint can be generalized to other background knowledge.

## 8 ACKNOWLEDGMENTS

The authors are grateful to the anonymous reviewers for valuable comments and suggestions. Rui Song's research is partially supported by a grant from the National Science Foundation DMS-1555244.

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

## A  ADDITIONAL GRAPH TERMINOLOGY

Given the node set, the weighted DAG can be uniquely determined by its weighted adjacency matrix, i.e., there is a one-to-one transformation between $\mathcal{G}$ and $B$. Suppose the graph nodes $X$ in $\mathcal{G}$ are sorted in its topological order (corresponding to elementary transformation of the matrix), then the matrix $B$ is strictly upper triangular with the diagonal elements as 0. Therefore, for an identity matrix $I$ with the same dimension as $B$, $I - B^{\top}$ is invertible since all its diagonal elements are 1 (positive).

## B  ALGORITHM: ANOCE-CVAE

The first part of the ANOCE-CVAE algorithm is on learning causal DAG from the observational data in the constrained space, by minimizing the loss function in Equation 7 using blackbox stochastic optimization solvers. Here, to minimize the loss in 7 and satisfy both $h_1(B) = 0$ and $h_2(B) = 0$, we simultaneously update $\lambda_1$ and $\lambda_2$ and increase $c$ and $d$ to infinity, by modifying the basic technique in Yu et al. (2019), corresponding to Part One.II.A.b and Part One.II.B in Algorithm 1. The second part is to estimate causal effects in the ANOCE table from the learned causal structure, based on the results from Theorem 3.1. Here, we numerically calculate the natural indirect effect for mediator $IM$ based on Corollary F.1 in step IV of the second part.

---

**Algorithm 1** Analysis of Causal Effects via Constrained VAE (ANOCE-CVAE)

---

Global: Dataset $X = \{A, M, Y\}$, sample size $n$, dimension of mediators $p$, max iteration $K$,
       number of epoch $H$, original learning rate $r_0$, tolerance of constraint to zero $\delta$,
       parameter update bound $U$, tuning parameters $\rho$ and $\omega$, and penalty terms $c$ and $d$;
Local: mean and standard variance of $\epsilon$ $\mu_\epsilon$ and $\sigma_\epsilon$, mean and standard variance of $X$ $\mu_X$ and $\sigma_X$,
       weights in multilayer perceptrons of encoder and decoder $\theta = \{W^{(1)}, W^{(2)}, W^{(3)}, W^{(4)}\}$,
       Lagrange multipliers $\lambda_1$ and $\lambda_2$, penalty terms $c$ and $d$, $(p + 2) \times (p + 2)$ matrix $B$,
       Loss function $L$, old and new values of the first constraint $h_1^{\text{old}}$ and $h_1^{\text{new}}$,
       old and new values of the second constraint $h_2^{\text{old}}$ and $h_2^{\text{new}}$, and learning rate $r$;
Output: estimated matrix $\widehat{B}$, total effect $TE$, natural direct and indirect effect
       $DE$ and $IE$, and natural direct and indirect effect for mediator $DM$ and $IM$.

Part One: Generate matrix $\widehat{B}$ via Constrained Variational Auto-Encoder;
 I. Initialization: $\lambda_1 \leftarrow 0$; $\lambda_2 \leftarrow 0$; $c \leftarrow 1$; $d \leftarrow 1$; $r \leftarrow r_0$; $B = \mathbf{0}_{(p+2)\times(p+2)}$; $h_1^{\text{old}} \leftarrow \infty$; $h_2^{\text{old}} \leftarrow \infty$;
 II. For step $k$, $k = 1, \cdots, K$:
    A. While $c \times d < U$ :
       a). For epoch $i$, $i = 1, \cdots, H$:
         1. Build Encoder $(\mu_\epsilon, \sigma_\epsilon) \leftarrow (I_{p+2} - B^{\top})MLP\{X, W^{(1)}, W^{(2)}\}$;
         2. Build Decoder $(\mu_X, \sigma_X) \leftarrow MLP\{(I_{p+2} - B^{\top})^{-1}\epsilon, W^{(3)}, W^{(4)}\}$;
         3. Calculate values of constraints $h_1^{\text{new}} \leftarrow h_1(B)$ and $h_2^{\text{new}} \leftarrow h_2(B)$,
           and the loss function $L \leftarrow L_{c,d}(B, W^{(1)}, W^{(2)}, W^{(3)}, W^{(4)}, \lambda_1, \lambda_2)$;
         4. Use backward to update parameters $\{B, W^{(1)}, W^{(2)}, W^{(3)}, W^{(4)}\}$;
         5. Update learning rate $r$;
       b). If $h_1^{\text{new}} > \rho h_1^{\text{old}}$ and $h_2^{\text{new}} > \rho h_2^{\text{old}}$: $c \leftarrow c \times \omega$; $d \leftarrow d \times \omega$;
         Elseif $h_1^{\text{new}} > \rho h_1^{\text{old}}$ and $h_2^{\text{new}} < \rho h_2^{\text{old}}$: $c \leftarrow c \times \omega$;
         Elseif $h_1^{\text{new}} < \rho h_1^{\text{old}}$ and $h_2^{\text{new}} > \rho h_2^{\text{old}}$: $d \leftarrow d \times \omega$;
         Else: Break;
    B. $h_1^{\text{old}} \leftarrow h_1^{\text{new}}$; $h_2^{\text{old}} \leftarrow h_2^{\text{new}}$; $\lambda_1 \leftarrow \lambda_1 \times h_1^{\text{new}}$; $\lambda_2 \leftarrow \lambda_2 \times h_2^{\text{new}}$;
    C. If $h_1^{\text{new}} < \delta$ and $h_2^{\text{new}} < \delta$: Break;
 III. Output $\widehat{B} \leftarrow B$;

---

---

**Algorithm 2** ANOCE-CVAE (cont.)

---

Part Two: Estimate causal effects in ANOCE based on matrix $\widehat{B}$;

   I. According to Equation 3:
     A. Get $\widehat{\gamma}$ as the direct effect $DE$;
     B. Get $\widehat{\alpha}$ as the effect of $A$ on $M$, $\widehat{\beta}$, and the inside matrix $\widehat{B}_M$;
   II. Get $\widehat{\zeta} \equiv (I_p - B_M^\top)^{-1}\widehat{\alpha}$ that represents the causal effect of $A$ on $M$;
   III. Get $\widehat{\beta}^\top \widehat{\zeta}$ that represents the total natural indirect effect $IE$;
     For each mediator $M_i$, $i = 1, \cdots, p$: Define the natural direct effect for $M_i$ as $DM[i] = \widehat{\alpha}[i]\widehat{\zeta}[i]$;
   IV. Get the natural indirect effect for mediator:
     For each mediator $M_i$, $i = 1, \cdots, p$:
     A. Delete $M_i$ from the matrix $\widehat{B}$ and get $\widehat{B'_i}$;
     B. Repeat step II. with reduced matrix $\widehat{B'_i}$ and get $\widehat{\beta'}$ and $\widehat{\zeta'}$;
     C. Calculate the effect difference as the total mediation effect $\widehat{\beta}^\top \widehat{\zeta} - \widehat{\beta'}^\top \widehat{\zeta'}$
     D. Define the natural indirect effect for $M_i$ as $IM[i] = \{\widehat{\beta}^\top \widehat{\zeta} - \widehat{\beta'}^\top \widehat{\zeta'}\} - DM[i]$;
   V. Define the total effect $TE = \widehat{\gamma} + \widehat{\beta}^\top \widehat{\zeta}$.

---

## C  ADDITIONAL SIMULATION STUDIES

In this section, we give more details on simulation studies to investigate the finite sample performance of the proposed method for learning causal effects with multiple mediators, in comparison to the popular causal discovery methods, including the PC, the ICA-LiNGAM, the NOTEARS, and the DAG-GNN. The computing infrastructure used is a virtual machine in the compute engine of Google Cloud Platform with 8 processor cores and 32GB memory. The average runtime for each result is around 1 to 2 hours.

### C.1  DATA GENERATION

We first generate a random DAG from the Erdős-Reńyi (ER) or the Scale-Free (SF) network (Barabási & Albert, 1999) with an expected node degree. Then, we remove all in-edges (from precedent nodes) of the first node as $A$ and remove all out-edges (from descendent nodes) of the last node as $Y$, and thus, the remaining nodes are the mediators $M$. Edges in DAGs for all scenarios are randomly assigned with weights ($w \in \{-1, 1\}$ with equal probability) to obtain the weighted adjacency matrix $B$. Specifically, the true DAGs in Scenarios 3 and 4 are generated from the Erdős-Reńyi (ER) model with an expected degree as 2, where we set number of nodes $d = 12$ (i.e. $p = 10$) in Scenario 3 and $d = 32$ (i.e. $p = 30$) in Scenario 4. Note that we consider fully identifiable models in Section 5.1 so that it is meaningful to evaluate causal effects from the estimated graph. In Section 5.2, we repeat the above generation procedure with $d = 12$ to generate the true graph from both the ER and the Scale-Free (SF) networks with the expected degree as 1, 2, and 4, denoted as Cases ER1, ER2, ER4, SF1, SF2, and SF4, respectively. Here, to be consistent with Section 5.1, we refer Scenario 3 (generated by the ER with the degree as 2) as Case ER2.

The synthetic datasets $\{A, M, Y\}$ are generated from Model 1, where the error variables in $\epsilon \equiv [\epsilon_A, \epsilon_M^\top, \epsilon_Y]^\top$ independently follow a normal distribution with mean 0 and noise 0.5 except for the binary exposure in Scenario $4^*$. Here, we add a baseline of 1.0 on the outcome $Y$. Note that the Gaussian exposure in Scenario 4 and the binary exposure in Scenario $4^*$ have the same mean and noise and thus their results are comparable.

### C.2  IMPLEMENTATION DETAILS

We detail the implementation for the proposed ANOCE-CVAE and comparison partners as follows:

• **ANOCE-CVAE:** The ANOCE-CVAE is implemented based on PyTorch (Paszke et al., 2017), using Adam (Kingma & Ba, 2014) to minimize the loss function in Equation 7. We set the batch

size as 25 for $n = 50$ and 100 for $n = 500$ with hidden nodes as $p^2$, the initial learning rate as 0.003 with an update rule as $r \leftarrow r/\{log(c) + log(d) + 0.01\}$ where $c$ and $d$ are penalty terms for two constraints, and the parameter update bound as $U = 10^{20}$, for all settings. Following the recommendation of Yu et al. (2019), we find that their tuned parameters $\rho = 0.25$ and $\omega = 10$ also work well in our settings, and we adopt the Huber-norm regularization of $B$ for a better convergence. Here, the variational posterior and the likelihood are parameterized as Gaussian with unit noise to approximate the underlying true model. The code is publicly available at an anonymous repository at `https://github.com/anoce-cvae/ANOCE-CVAE`.

- **PC (Spirtes et al., 2000):** We set the Fisher-z test for the PC algorithm with the p-value as 0.01 for all settings. The implementation is available through the py-causal package at `https://github.com/bd2kccd/py-causal`, written in highly optimized Java codes. Also see examples here `https://github.com/bd2kccd/py-causal/blob/development/example/py-causal%20-%20PC-ALL%20in%20Action.ipynb`.

- **ICA-LiNGAM (Shimizu et al., 2006):** The ICA-LiNGAM assumes linear non-Gaussian additive model to recover the weighted adjacency matrix. We implement the ICA-LiNGAM with default hyper-parameters through the lingam package for all settings. See their repository at `https://github.com/cdt15/lingam`.

- **NOTEARS (Zheng et al., 2018):** The NOTEARS estimates the weighted adjacency matrix by formulating the optimization with an acyclicity constraint. The implementation is available at their repository at `https://github.com/xunzheng/notears`. We set the loss function as the least square error with the $L_1$ regularization. We find the NOTEARS is sensitive to the choice of the $L_1$ regularization in our settings. For a fair comparison, we set the $L_1$ penalty parameter as 0.03 (instead of the default 0.1) for all settings, which achieves an overall good performance in most cases. Note the author modified their acyclicity constraint in their codes to be the one used in Yu et al. (2019) (i.e. Equation 4). We also use the same acyclicity constraint for NOTEARS, DAGGNN, and our method for a fair comparison. Other hyper-parameters are set as default in their repository.

- **DAG-GNN (Yu et al., 2019):** The DAG-GNN incorporates the variational auto-encoder into causal discovery with a modified smooth characterization on acyclicity in the evidence lower bound as the loss function. Codes are available at their repository at `https://github.com/fishmoon1234/DAG-GNN` based on PyTorch (Paszke et al., 2017). We set the same hyper-parameters used in our ANOCE-CVAE for a fair comparison. Specifically, we use Adam (Kingma & Ba, 2014) to minimize the loss function, and set the batch size as 25 for $n = 50$ and 100 for $n = 500$ with hidden nodes as $p^2$. The initial learning rate is set as 0.003 with an update rule as $r \leftarrow r/\{log(c_0) + 0.01\}$ where $c_0$ is penalty term for the acyclicity constraint. The rest settings are the same as the default in their codes.

In the comparison studies (see Section 5.2 and C.4), we use a uniform graph threshold as 0.3 (commonly used in current literature) for all algorithms to prune the noise edges for a fair comparison. In addition, we also provide the results under the graph threshold as 0.4 for additional comparison.

## C.3 ADDITIONAL RESULTS OF ANOCE-CVAE

In this section, we provide additional simulation results for the ANOCE-CVAE. Following Section 5.1, the numerical results are summarized in Table 2 (for Scenario 1 to 3) and Table 3 (for Scenario 4 and $4^*$), including the bias of the estimated $TE$, $DE$, $IE$, $DM$ and $IM$ for each mediator with their standard error, over 100 replications. Note that due to limited space, we save the numerical results of the $IM$ in Table 3.

From the results in Table 2, it is clear that the estimated $TE$, $DE$, $IE$, $DM$ and $IM$ for each mediator are close to the true values as the sample size increases in Scenario 1 to 3, which indicates the good performance of our proposed method on identifying the causal effects regardless of the sparsity. With the expected node degree increasing, one can observe a slightly larger bias and standard error of the estimated causal effects as expected, as shown in Table 2. Based on Table 3, the results of Scenario 4 and Scenario $4^*$ are merely identical under different sample sizes, indicating our proposed method can handle either discrete or continuous exposure. In addition, by comparing the results of Scenario 3 and 4 where we fix the expected node degree as 2, one can observe a slightly larger bias

of the estimated causal effects but of a similar small scale, as the dimension of mediators $p$ increases, which implies the stability of our method under the high-dimensional setting.

Table 2: The bias and standard error of the estimated causal effects for Scenario 1 to 3 ($p = 10$).

| Results | Scen. 1 (simplest) | | Scen. 2 (parallel) | | Scen. 3 (interacted) | |
|---|---|---|---|---|---|---|
| | $n = 50$ | $n = 500$ | $n = 50$ | $n = 500$ | $n = 50$ | $n = 500$ |
| $TE$ | 0.19 (0.02) | 0.01 (0.00) | 0.66 (0.05) | 0.03 (0.02) | 0.15 (0.05) | 0.05 (0.03) |
| $DE$ | 0.23 (0.02) | 0.01 (0.00) | 0.78 (0.02) | 0.12 (0.06) | 0.78 (0.03) | 0.15 (0.06) |
| $IE$ | 0.04 (0.01) | 0.00 (0.00) | 1.44 (0.04) | 0.08 (0.06) | 0.63 (0.04) | 0.20 (0.04) |
| $DM_1$ | 0.01 (0.00) | 0.00 (0.00) | 0.69 (0.03) | 0.02 (0.01) | 0.48 (0.04) | 0.06 (0.02) |
| $DM_2$ | 0.00 (0.00) | 0.00 (0.00) | 0.43 (0.03) | 0.01 (0.01) | 0.00 (0.00) | 0.00 (0.00) |
| $DM_3$ | 0.00 (0.00) | 0.00 (0.00) | 0.65 (0.02) | 0.01 (0.02) | 0.00 (0.00) | 0.00 (0.00) |
| $DM_4$ | 0.00 (0.00) | 0.00 (0.00) | 0.65 (0.03) | 0.02 (0.01) | 0.20 (0.02) | 0.06 (0.01) |
| $DM_5$ | 0.01 (0.00) | 0.00 (0.00) | 0.68 (0.03) | 0.02 (0.01) | 0.44 (0.02) | 0.10 (0.01) |
| $DM_6$ | 0.01 (0.00) | 0.00 (0.00) | 0.67 (0.02) | 0.02 (0.02) | 0.27 (0.04) | 0.03 (0.01) |
| $DM_7$ | 0.00 (0.00) | 0.00 (0.00) | 0.62 (0.03) | 0.03 (0.01) | 0.67 (0.03) | 0.16 (0.03) |
| $DM_8$ | 0.00 (0.00) | 0.00 (0.00) | 0.66 (0.03) | 0.01 (0.01) | 0.59 (0.03) | 0.14 (0.03) |
| $DM_9$ | 0.00 (0.00) | 0.00 (0.00) | 0.65 (0.02) | 0.01 (0.01) | 0.00 (0.00) | 0.00 (0.00) |
| $DM_{10}$ | 0.01 (0.00) | 0.00 (0.00) | 0.66 (0.03) | 0.01 (0.01) | 0.10 (0.02) | 0.05 (0.01) |
| $IM_1$ | 0.00 (0.00) | 0.00 (0.00) | 0.08 (0.01) | 0.08 (0.01) | 0.76 (0.03) | 0.16 (0.04) |
| $IM_2$ | 0.00 (0.00) | 0.00 (0.00) | 0.09 (0.01) | 0.01 (0.01) | 0.00 (0.00) | 0.00 (0.00) |
| $IM_3$ | 0.00 (0.00) | 0.00 (0.00) | 0.03 (0.01) | 0.03 (0.01) | 0.00 (0.00) | 0.00 (0.00) |
| $IM_4$ | 0.00 (0.00) | 0.00 (0.00) | 0.03 (0.01) | 0.05 (0.01) | 0.16 (0.01) | 0.03 (0.01) |
| $IM_5$ | 0.00 (0.00) | 0.00 (0.00) | 0.06 (0.01) | 0.12 (0.01) | 0.02 (0.01) | 0.03 (0.00) |
| $IM_6$ | 0.00 (0.00) | 0.00 (0.00) | 0.03 (0.01) | 0.04 (0.01) | 0.66 (0.02) | 0.09 (0.02) |
| $IM_7$ | 0.00 (0.00) | 0.00 (0.00) | 0.09 (0.01) | 0.00 (0.01) | 0.05 (0.02) | 0.04 (0.01) |
| $IM_8$ | 0.00 (0.00) | 0.00 (0.00) | 0.03 (0.01) | 0.02 (0.01) | 0.11 (0.02) | 0.06 (0.01) |
| $IM_9$ | 0.00 (0.00) | 0.00 (0.00) | 0.03 (0.01) | 0.01 (0.01) | 0.00 (0.00) | 0.00 (0.00) |
| $IM_{10}$ | 0.00 (0.00) | 0.00 (0.00) | 0.07 (0.01) | 0.08 (0.01) | 0.03 (0.01) | 0.00 (0.00) |

Table 3: The bias and standard error of the estimated causal effects for Scenario 4 and $4^*$ ($p = 30$).

| Results | Scen. 4 (continuous) | | Scen. $4^*$ (binary) | |
|---|---|---|---|---|
| | $n = 50$ | $n = 500$ | $n = 50$ | $n = 500$ |
| $TE$ | 0.29 (0.04) | 0.20 (0.02) | 0.09 (0.03) | 0.05 (0.01) |
| $DE$ | 0.90 (0.01) | 0.28 (0.02) | 0.85 (0.01) | 0.25 (0.02) |
| $IE$ | 1.19 (0.04) | 0.48 (0.03) | 0.94 (0.03) | 0.30 (0.02) |
| $DM_1$ | 0.12 (0.02) | 0.08 (0.01) | 0.18 (0.02) | 0.07 (0.01) |
| $DM_2$ | 0.61 (0.02) | 0.02 (0.01) | 0.45 (0.02) | 0.02 (0.01) |
| $DM_3$ | 0.00 (0.01) | 0.03 (0.01) | 0.01 (0.01) | 0.02 (0.01) |
| $DM_4$ | 0.31 (0.03) | 0.25 (0.02) | 0.09 (0.02) | 0.07 (0.01) |
| $DM_5$ | 0.01 (0.01) | 0.07 (0.01) | 0.05 (0.03) | 0.10 (0.02) |
| $DM_6$ | 0.00 (0.00) | 0.00 (0.00) | 0.00 (0.00) | 0.00 (0.00) |
| $DM_7$ | 0.37 (0.02) | 0.19 (0.02) | 0.60 (0.04) | 0.18 (0.02) |
| $DM_8$ | 0.02 (0.01) | 0.00 (0.01) | 0.05 (0.01) | 0.02 (0.02) |
| $DM_9$ | 0.00 (0.00) | 0.00 (0.00) | 0.00 (0.00) | 0.00 (0.00) |
| $DM_{10}$ | 0.02 (0.01) | 0.09 (0.01) | 0.14 (0.01) | 0.15 (0.01) |
| $DM_{11}$ | 0.00 (0.01) | 0.00 (0.00) | 0.01 (0.01) | 0.00 (0.00) |
| $DM_{12}$ | 0.02 (0.00) | 0.01 (0.01) | 0.06 (0.01) | 0.01 (0.01) |
| $DM_{13}$ | 0.25 (0.02) | 0.11 (0.02) | 0.51 (0.04) | 0.16 (0.03) |
| $DM_{14}$ | 0.02 (0.01) | 0.10 (0.01) | 0.09 (0.02) | 0.16 (0.01) |
| $DM_{15}$ | 0.00 (0.00) | 0.00 (0.00) | 0.00 (0.00) | 0.00 (0.00) |
| $DM_{16}$ | 0.09 (0.01) | 0.10 (0.02) | 0.18 (0.03) | 0.13 (0.02) |
| $DM_{17}$ | 0.03 (0.01) | 0.07 (0.02) | 0.04 (0.03) | 0.06 (0.03) |
| $DM_{18}$ | 0.01 (0.00) | 0.00 (0.00) | 0.00 (0.00) | 0.00 (0.00) |
| $DM_{19}$ | 0.02 (0.00) | 0.01 (0.00) | 0.00 (0.00) | 0.00 (0.00) |
| $DM_{20}$ | 0.01 (0.00) | 0.00 (0.00) | 0.08 (0.01) | 0.01 (0.01) |
| $DM_{21}$ | 0.00 (0.00) | 0.00 (0.00) | 0.00 (0.00) | 0.00 (0.00) |
| $DM_{22}$ | 0.00 (0.00) | 0.00 (0.00) | 0.00 (0.00) | 0.00 (0.00) |
| $DM_{23}$ | 0.00 (0.00) | 0.00 (0.00) | 0.00 (0.00) | 0.00 (0.00) |
| $DM_{24}$ | 0.00 (0.00) | 0.00 (0.00) | 0.00 (0.00) | 0.00 (0.00) |
| $DM_{25}$ | 0.00 (0.00) | 0.00 (0.00) | 0.00 (0.00) | 0.00 (0.00) |
| $DM_{26}$ | 0.00 (0.00) | 0.00 (0.00) | 0.00 (0.00) | 0.00 (0.00) |
| $DM_{27}$ | 0.00 (0.00) | 0.00 (0.00) | 0.00 (0.00) | 0.00 (0.00) |
| $DM_{28}$ | 0.01 (0.01) | 0.02 (0.01) | 0.01 (0.01) | 0.01 (0.00) |
| $DM_{29}$ | 0.00 (0.00) | 0.00 (0.00) | 0.00 (0.00) | 0.00 (0.00) |
| $DM_{30}$ | 0.00 (0.00) | 0.00 (0.00) | 0.00 (0.00) | 0.00 (0.00) |

## C.4 ADDITIONAL COMPARISON STUDIES

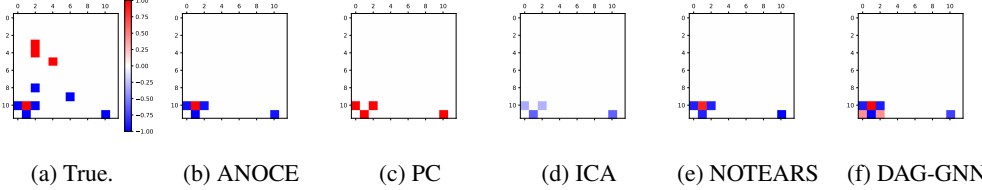

(a) True.    (b) ANOCE    (c) PC    (d) ICA    (e) NOTEARS    (f) DAG-GNN

Figure 5: The averaged estimated $\widehat{B}^{\top}$ for Case ER1 under different methods with threshold 0.3.

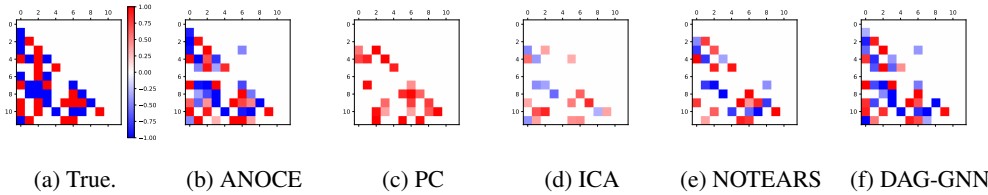

(a) True.    (b) ANOCE    (c) PC    (d) ICA    (e) NOTEARS    (f) DAG-GNN

Figure 6: The averaged estimated $\widehat{B}^{\top}$ for Case ER4 under different methods with threshold 0.3.

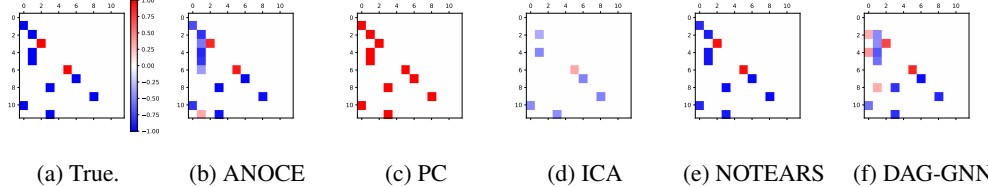

(a) True.    (b) ANOCE    (c) PC    (d) ICA    (e) NOTEARS    (f) DAG-GNN

Figure 7: The averaged estimated $\widehat{B}^{\top}$ for Case SF1 under different methods with threshold 0.3.

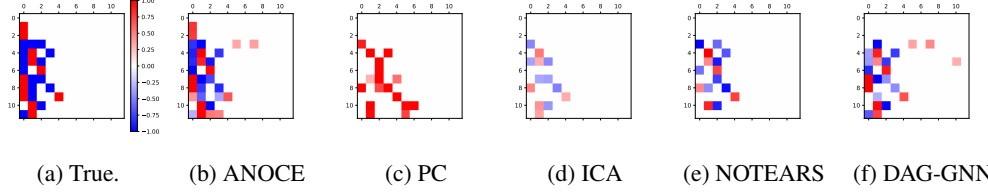

(a) True.    (b) ANOCE    (c) PC    (d) ICA    (e) NOTEARS    (f) DAG-GNN

Figure 8: The averaged estimated $\widehat{B}^{\top}$ for Case SF2 under different methods with threshold 0.4.

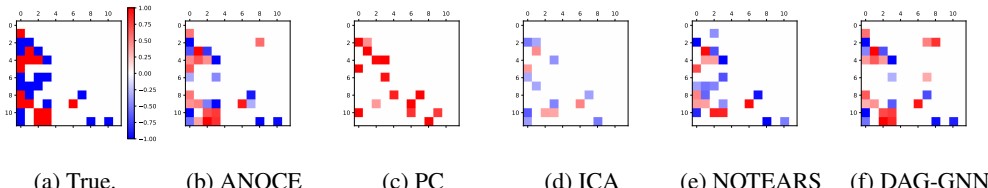

(a) True.    (b) ANOCE    (c) PC    (d) ICA    (e) NOTEARS    (f) DAG-GNN

Figure 9: The averaged estimated $\widehat{B}^{\top}$ for Case SF4 under different methods with threshold 0.4.

This section provides more results on comparison studies against the existing methods. The data generation and implementation details are provided in Section C.1 and C.2. Following Section 5.2

Table 4: The empirical comparison results under different causal discovery methods with the graph threshold as 0.3.

| Methods | Case | ER1 | ER2 | ER4 | SF1 | SF2 | SF4 |
|---|---|---|---|---|---|---|---|
| ANOCE-CVAE | FDR | 0.00 (0.14) | 0.00 (0.09) | 0.14 (0.03) | 0.21 (0.06) | 0.17 (0.05) | 0.13 (0.06) |
| | TPR | 0.50 (0.00) | 1.00 (0.08) | 0.93 (0.05) | 1.00 (0.03) | 0.96 (0.05) | 0.81 (0.06) |
| | SHD | 5.00 (1.01) | 0.00 (3.57) | 8.00 (2.44) | 3.00 (1.49) | 6.00 (2.15) | 10.00 (2.65) |
| PC | FDR | 0.00 (0.10) | 0.50 (0.04) | 0.23 (0.05) | 0.00 (0.01) | 0.29 (0.04) | 0.27 (0.05) |
| | TPR | 0.40 (0.01) | 0.26 (0.01) | 0.41 (0.04) | 1.00 (0.00) | 0.46 (0.02) | 0.34 (0.02) |
| | SHD | 6.00 (0.55) | 19.00 (0.51) | 29.00 (2.01) | 0.00 (0.17) | 19.00 (1.19) | 25.00 (1.18) |
| ICA-LiNGAM | FDR | 0.00 (0.18) | 0.08 (0.16) | 0.15 (0.12) | 0.00 (0.14) | 0.00 (0.18) | 0.00 (0.14) |
| | TPR | 0.40 (0.12) | 0.52 (0.14) | 0.41 (0.10) | 0.64 (0.17) | 0.50 (0.17) | 0.47 (0.13) |
| | SHD | 6.00 (1.50) | 12.00 (4.08) | 26.00 (4.67) | 4.00 (2.98) | 13.00 (6.29) | 17.00 (5.36) |
| NOTEARS | FDR | 0.00 (0.02) | 0.00 (0.06) | 0.04 (0.06) | 0.00 (0.00) | 0.00 (0.07) | 0.04 (0.03) |
| | TPR | 0.50 (0.00) | 0.78 (0.09) | 0.63 (0.08) | 1.00 (0.00) | 0.58 (0.08) | 0.72 (0.07) |
| | SHD | 5.00 (0.10) | 5.00 (2.29) | 15.00 (3.90) | 0.00 (0.00) | 11.00 (2.82) | 9.00 (2.59) |
| DAG-GNN | FDR | 0.29 (0.07) | 0.15 (0.06) | 0.13 (0.04) | 0.29 (0.07) | 0.13 (0.06) | 0.11 (0.05) |
| | TPR | 0.50 (0.00) | 0.74 (0.06) | 0.80 (0.06) | 0.91 (0.06) | 0.77 (0.07) | 0.75 (0.05) |
| | SHD | 7.00 (0.49) | 9.00 (2.22) | 12.00 (3.08) | 5.00 (1.66) | 9.00 (2.23) | 10.00 (1.71) |

and C.2, we use a graph thresholds as 0.3 (commonly used in current literature) or 0.4 (for additional comparison) in all algorithms to prune the noise edges for a fair comparison.

The estimated graphs (after pruning) are evaluated by three metrics: the false discovery rate (FDR), the true positive rate (TPR), and the structural Hamming distance (SHD, the smallest number of edge additions, deletions, and reversals to convert the estimated graph into the true DAG). Here, the SHD takes into account both false positives and negatives and a lower SHD indicates a better estimate of the causal graph. The FDR, TPR, and SHD of the averaged estimated matrix $\widehat{B}^{\top}$ with their standard deviation over 100 replications are reported in Table 4 for the graph threshold as 0.3 and in Table 5 for the graph threshold as 0.4, under different methods for all six cases with sample size $n = 500$.

Table 5: The empirical comparison results under different causal discovery methods with the graph threshold as 0.4.

| Methods | Case | ER1 | ER2 | ER4 | SF1 | SF2 | SF4 |
|---|---|---|---|---|---|---|---|
| ANOCE-CVAE | FDR | 0.00 (0.13) | 0.00 (0.08) | 0.07 (0.04) | 0.08 (0.05) | 0.08 (0.04) | 0.07 (0.05) |
| | TPR | 0.50 (0.00) | 1.00 (0.10) | 0.90 (0.05) | 1.00 (0.03) | 0.92 (0.05) | 0.78 (0.07) |
| | SHD | 5.00 (0.92) | 0.00 (3.52) | 6.00 (2.91) | 1.00 (0.90) | 4.00 (1.89) | 9.00 (2.62) |
| PC | FDR | 0.00 (0.10) | 0.40 (0.04) | 0.20 (0.05) | 0.00 (0.01) | 0.31 (0.04) | 0.27 (0.05) |
| | TPR | 0.40 (0.01) | 0.26 (0.01) | 0.39 (0.04) | 1.00 (0.00) | 0.42 (0.02) | 0.34 (0.02) |
| | SHD | 6.00 (0.55) | 19.00 (0.51) | 29.00 (2.01) | 0.00 (0.17) | 20.00 (1.19) | 25.00 (1.18) |
| ICA-LiNGAM | FDR | 0.00 (0.30) | 0.00 (0.15) | 0.08 (0.12) | 0.00 (0.15) | 0.00 (0.19) | 0.00 (0.15) |
| | TPR | 0.20 (0.14) | 0.39 (0.14) | 0.29 (0.09) | 0.45 (0.16) | 0.27 (0.17) | 0.22 (0.13) |
| | SHD | 8.00 (1.72) | 14.00 (3.90) | 30.00 (4.28) | 6.00 (2.51) | 19.00 (5.48) | 25.00 (4.99) |
| NOTEARS | FDR | 0.00 (0.00) | 0.00 (0.04) | 0.04 (0.05) | 0.00 (0.00) | 0.00 (0.06) | 0.05 (0.03) |
| | TPR | 0.50 (0.00) | 0.65 (0.08) | 0.59 (0.07) | 1.00 (0.00) | 0.58 (0.09) | 0.56 (0.08) |
| | SHD | 5.00 (0.00) | 8.00 (1.86) | 17.00 (3.54) | 0.00 (0.00) | 11.00 (2.83) | 14.00 (2.72) |
| DAG-GNN | FDR | 0.29 (0.09) | 0.15 (0.05) | 0.11 (0.05) | 0.17 (0.05) | 0.06 (0.06) | 0.09 (0.03) |
| | TPR | 0.50 (0.00) | 0.74 (0.07) | 0.78 (0.07) | 0.91 (0.07) | 0.65 (0.08) | 0.66 (0.06) |
| | SHD | 7.00 (0.67) | 9.00 (1.64) | 12.00 (3.36) | 3.00 (1.08) | 10.00 (2.32) | 12.00 (1.71) |

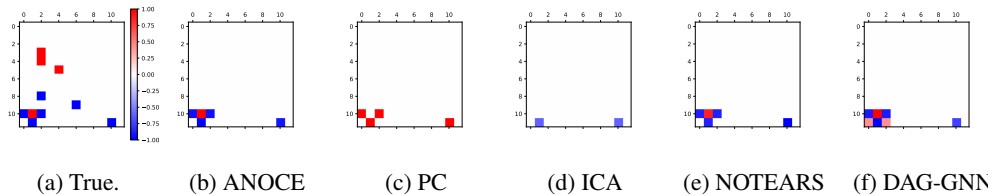

(a) True.  (b) ANOCE  (c) PC  (d) ICA  (e) NOTEARS  (f) DAG-GNN

Figure 10: The averaged estimated $\widehat{B}^{\top}$ for Case ER1 under different methods with threshold $0.4$.

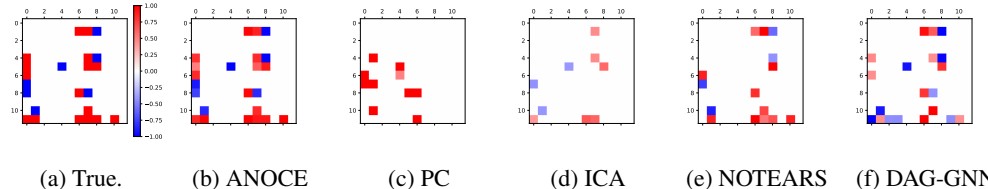

(a) True.  (b) ANOCE  (c) PC  (d) ICA  (e) NOTEARS  (f) DAG-GNN

Figure 11: The averaged estimated $\widehat{B}^{\top}$ for Case ER2 under different methods with threshold $0.4$.

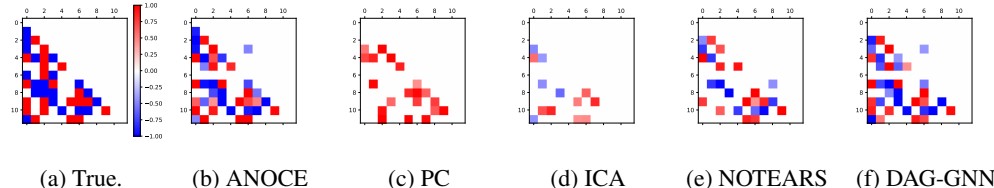

(a) True.  (b) ANOCE  (c) PC  (d) ICA  (e) NOTEARS  (f) DAG-GNN

Figure 12: The averaged estimated $\widehat{B}^{\top}$ for Case ER4 under different methods with threshold $0.4$.

Besides Figure 3 for Case ER2 with a graph threshold as 0.3 shown in the main text, we also illustrate the averaged estimated matrix $\widehat{B}^{\top}$ over 100 replications under different methods for Case ER1, ER4,

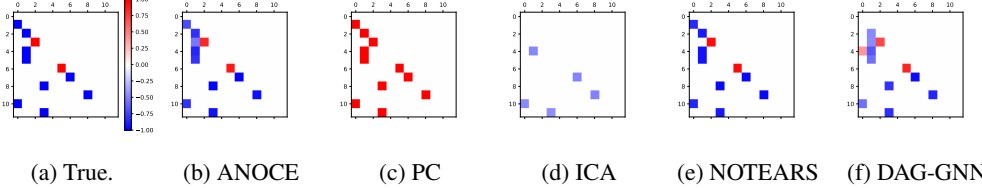

(a) True.  (b) ANOCE  (c) PC  (d) ICA  (e) NOTEARS  (f) DAG-GNN

Figure 13: The averaged estimated $\widehat{B}^\top$ for Case SF1 under different methods with threshold $0.4$.

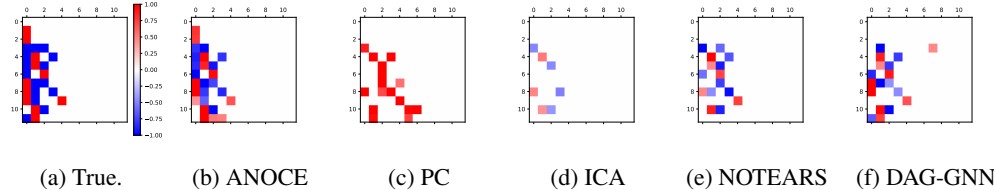

(a) True.  (b) ANOCE  (c) PC  (d) ICA  (e) NOTEARS  (f) DAG-GNN

Figure 14: The averaged estimated $\widehat{B}^\top$ for Case SF2 under different methods with threshold $0.4$.

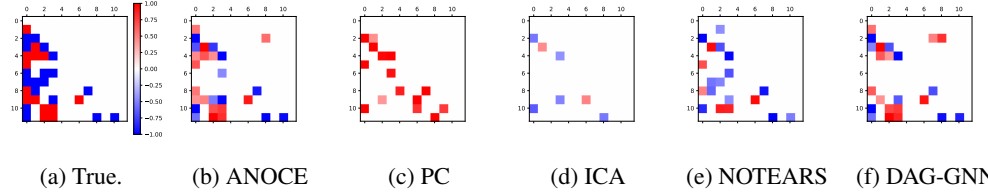

(a) True.  (b) ANOCE  (c) PC  (d) ICA  (e) NOTEARS  (f) DAG-GNN

Figure 15: The averaged estimated $\widehat{B}^\top$ for Case SF4 under different methods with threshold $0.4$.

SF1, SF2, and SF4 with a graph threshold as 0.3 in Figure 5, 6, 7, 8, 9, respectively, and for Case ER1, ER2, ER4, SF1, SF2, and SF4 with a graph threshold as 0.4 in Figure 10, 11, 12, 13, 14, 15, respectively, under $n = 500$.

From Table 4 and 5, it is clear that our algorithm performs the best among the five methods in most cases, followed by the other two score-based methods, i.e. the NOTEARS and the DAG-GNN. While the traditional methods (the PC and the ICA-LiNGAM) perform the worst with large SHD and small TPR. This finding supports the choice of the extension on the score-based method. Moreover, by comparing our performance with the DAG-GNN, one can observe a substantial gain in terms of the SHD and the TPR in most cases with comparable FDR. This validates the improvement of our method over the DAG-GNN by introducing the background knowledge in the causal discovery. Another supports are illustrated in Figure 3, 5, 6, 7, 8, 9, 10, 11, 12, 13, 14, and 15 for different settings, where the averaged estimated matrix $\widehat{B}^\top$ under the ANOCE-CVAE is approximately the same as the ground true graph $B^\top$ when $n = 500$. However, the PC and the ICA-LiNGAM can hardly recognize the true causal pattern. In addition, methods have a slightly better performance in terms of FDR and SHD while a slightly worse performance in terms of TPR under the graph threshold as 0.4, in comparison to the results under the graph threshold as 0.3.

# D  ADDITIONAL REAL DATA RESULTS

In this section, we provide additional real data analysis on the COVID-19.

## D.1  DATA COLLECTION

To better characterize the causality of the virus spreading under the Hubei lockdowns in China, we assume: 1) Hubei was the centre of the COVID-19 outbreak in China (Zhou et al., 2020); 2)

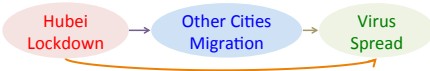

Figure 16: A direct acyclic graph illustrating the temporal causal relationship of the spreading network of the coronavirus outbreak under 2020 Hubei lockdowns in China.

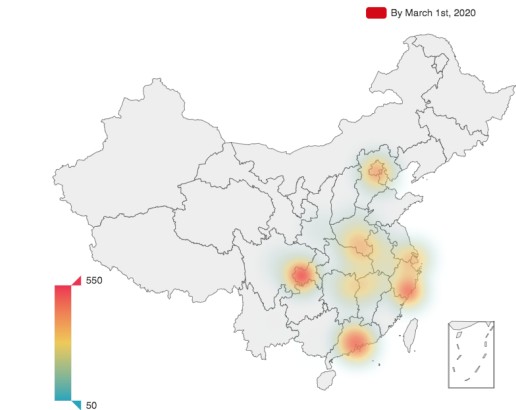

(a) The cumulative confirmed cases of selected cities.

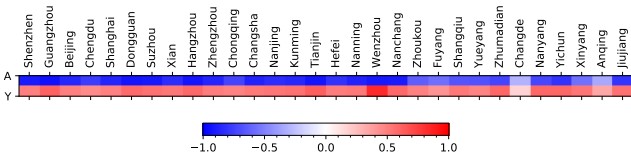

(b) The partial heat map of correlation matrix between cities and the treatment $A$ or the outcome $Y$.

Figure 17: Additional figures for the real data analysis of the COVID-19 spread.

the decreased migration outside Hubei was largely stimulated by the lockdown; 3) individual who departed to one destination would not return to the original departure due to travel restrictions in China. Under the above assumptions, it is reasonable to use a temporal causal relationship to describe the spread of COVID-19 under 2020 Hubei lockdowns as in Figure 16.

Next, we give more details on how to present components in Figure 16 with appropriate variables. First, we set the exposure $A$ as if Hubei is on lockdown, 0 for unlocked (before and on Jan 23rd) and 1 for locked (on and after Jan 24th). To select the candidate cities that contain most potential infected people, we rank cities outside Hubei by their received Wuhan (identified over 60% cases in China reported by the NHC) migration between Jan 1st, 2020 and Jan 22nd, 2020 (before the lockdown), and choose the top 30 cities (account for 69.17% of total Wuhan migration) as mediators $M$, to control the noise. We use the daily migration scale index (MSI) of each city as the value of each mediator, defined in Baidu Qianxi to describe the migration magnitude. By noticing the following facts: 1) it took usually 2 days to diagnose the COVID-19; 2) the estimated median incubation period is 5 days (Lauer et al., 2020); the outcome of interest $Y$ is defined as the daily increase rate of confirmed cases out of Hubei with a one-week (7=2+5) delay. We delete extreme data points (two outbreaks at jails), and the final dataset yields a total of 38 records.

## D.2 Initial Data Analysis

Figure 17a demonstrates the location of selected cities on the Chinese map with the color representing its cumulative confirmed cases by March 1st, 2020. It can be observed that the selected mediators are either located around Hubei province or are big central cities such as Beijing that has a large population and migration scale. In addition, we provide the partial heat map of the correlation matrix between cities and the treatment $A$ or the outcome $Y$ as illustrated in Figure 17b. Here, one could observe that cities' MSIs are highly positive correlated with the lockdown while are highly negative correlated with the daily increase rate of confirmed cases.

## D.3 Additional Results

Table 6 lists all the numerical facts of the selected cities, including the population (million), the cumulative migration scale index (MSI) during the data period (Jan 12th to Feb 20th, 2020), the ratio of received Wuhan migration between Jan 1st to Jan 22nd, 2020 (before the lockdown), the cumulative confirmed cases by March 1st, 2020, and cities' direct ($DM$) and indirect effect ($IM$). Note the selected 30 cities in Table 6 are sorted according to their cumulative MSI, and its order is used as the order of the mediators in this paper. It can be seen from Table 6 that the population, the cumulative MSI, the ratio of received Wuhan migration, and the confirmed cases are highly correlated among selected cities as expected. Note that we list these factors to assist the interpretation of the results, while none of them is used to estimate the DAG of interest. Therefore, similar values of factors don't necessarily imply similar causal effects.

Besides the general causal pattern of cities' estimated $DM$s and $IM$s stated in the main text, we provide more interpretation on the level of the individual city. Here, we compare three groups of cities that are of similar scale (population and migration) or geographic position, to specify our results. First, we compare the results between Beijing and Shanghai, where both cities have comparable population scale and are the center of politic or economic in China (the correlation between the MSI of Beijing and the outcome yields the same value as of Shanghai as 0.50). It can be seen from Table 6 that Beijing and Shanghai have similar positive indirect effects as 0.247 and 0.235, respectively, while Shanghai has a slightly higher direct effect on controlling the virus as -0.069, which is possibly due to the smaller MSI of Shanghai. Second, we compare three cities located at Guangdong province (see Figure 18a and 18b), including Shenzhen, Guangzhou, and Dongguan. All three cities have positive effects, among which Guangzhou yields the highest direct effect as 0.847, followed by Dongguan. The effect size of these cities agrees with their correlation coefficients with the outcome, where the correlation between Guangzhou and $Y$ is 0.61, followed by Dongguan as 0.58 and Shenzhen as 0.50. The last comparison is among cities in southeastern China (see Figure 18a and 18b), including Suzhou, Hangzhou, and Wenzhou, all of which have a negative direct effect on the virus control and a positive indirect effect for the virus spread. Here, Wenzhou achieves the largest absolute value of the negative direct effect as -0.650, which conforms to its strict local shelter in home order after the Hubei lockdowns, where the correlation between Wenzhou and the outcome achieves the highest value as 0.83.

We summarize all different sources of causal effects in Table 6 as the ANOCE table of 2020 Hubei lockdowns on reducing the COVID-19 spread in China. Note that due to cities' different levels of control measures on the coronavirus outside Hubei as well as other possible confounders, there are some inconsistency between cities' DMs and their cumulative MSIs. We leave the extension with confounders for further investigation. Further interpretation of the real data analysis requires domain experts.

One can refer to Figure 18a and 18b for cities' $DM$s and $IM$s on the Chinese map. To check the reasonability of our results, we plot the estimated weighted matrix in Figure 4a, where the first node (indexed 0) represents the Hubei lockdowns, the last node (indexed 31) is the daily increase rate out of Hubei, and the middle 30 nodes (indexed 1-30) correspond to 30 selected cities in Figure 4b. From Figure 4a, we can observe: • 1). The color of the first column is almost all blue, indicating that locking Hubei down can reduce the migration of selected cities; • 2). An approximate red upper triangular among first 20 nodes implies a migration trend with positive effects among central cities with large MSI, i.e. relatively smaller cities tend to have positive effects on other cities with relatively larger MSI; • 3). An approximate red lower triangular among last 10 nodes indicates a weaker migration trend with an opposite direction among non-central cities with small MSI; • 4). There is

Table 6: Analysis of causal effects of 2020 Hubei lockdowns on reducing the COVID-19 spread in China regulated by Chinese major cities outside Hubei.

| Source | | | | | $d.f.$ | Causal effects | |
|---|---|---|---|---|---|---|---|
| Direct effect from the Hubei lockdowns | | | | | 1 | $DE = -0.078$ | |
| Inirect effect via Chinese major cities outside Hubei | | | | | $p = 30$ | $IE = -0.419$ | |
| Index | City name | Population | Ratio of Wuhan | Confirmed Cases | Cumulative MSI | | $DM$ | $IM$ |
| 1 | Shenzhen | 10.36 | 3.02% | 418 | 275.42 | 1 | 0.021 | 0.002 |
| 2 | Guangzhou | 12.70 | 3.22% | 346 | 270.38 | 1 | 0.847 | 0.176 |
| 3 | Beijing | 23.00 | 5.50% | 414 | 260.02 | 1 | -0.003 | 0.247 |
| 4 | Chengdu | 14.05 | 2.57% | 143 | 233.62 | 1 | -0.201 | -0.004 |
| 5 | Shanghai | 23.02 | 4.09% | 338 | 232.80 | 1 | -0.069 | 0.235 |
| 6 | Dongguan | 8.22 | 1.14% | 99 | 211.99 | 1 | 0.257 | 0.003 |
| 7 | Suzhou | 10.47 | 1.42% | 87 | 171.46 | 1 | -0.481 | 0.553 |
| 8 | Xian | 10.00 | 1.87% | 120 | 136.56 | 1 | 0.015 | -0.089 |
| 9 | Hangzhou | 8.70 | 1.77% | 169 | 135.73 | 1 | -0.335 | 0.144 |
| 10 | Zhengzhou | 10.14 | 3.04% | 157 | 132.58 | 1 | -0.019 | 0.309 |
| 11 | Chongqing | 30.17 | 4.97% | 576 | 128.98 | 1 | 0.313 | 0.050 |
| 12 | Changsha | 7.04 | 5.21% | 242 | 126.78 | 1 | 0.399 | 0.222 |
| 13 | Nanjing | 8.00 | 1.78% | 93 | 103.87 | 1 | -0.210 | -0.100 |
| 14 | Kunming | 6.43 | 1.24% | 53 | 95.35 | 1 | 0.186 | -0.075 |
| 15 | Tianjin | 12.94 | 1.02% | 136 | 90.50 | 1 | -0.092 | -0.372 |
| 16 | Hefei | 5.70 | 2.11% | 174 | 87.28 | 1 | -0.182 | 0.249 |
| 17 | Nanning | 6.66 | 1.01% | 55 | 76.63 | 1 | 0.114 | 0.152 |
| 18 | Wenzhou | 9.12 | 1.02% | 504 | 60.38 | 1 | -0.650 | 0.200 |
| 19 | Nanchang | 5.04 | 2.15% | 230 | 46.95 | 1 | -0.109 | 0.095 |
| 20 | Zhoukou | 8.95 | 1.50% | 76 | 42.01 | 1 | 0.045 | -0.043 |
| 21 | Fuyang | 7.60 | 1.25% | 155 | 39.26 | 1 | 0.031 | -0.023 |
| 22 | Shangqiu | 7.36 | 1.12% | 91 | 35.48 | 1 | 0.030 | -0.057 |
| 23 | Yueyang | 5.48 | 2.31% | 156 | 29.56 | 1 | 0.006 | -0.050 |
| 24 | Zhumadian | 7.23 | 2.34% | 139 | 28.86 | 1 | -0.070 | -0.075 |
| 25 | Changde | 5.72 | 1.05% | 82 | 28.49 | 1 | 0.002 | -0.003 |
| 26 | Nanyang | 10.26 | 1.91% | 156 | 27.71 | 1 | -0.039 | -0.106 |
| 27 | Yichun | 5.42 | 0.90% | 106 | 24.91 | 1 | -0.051 | -0.085 |
| 28 | Xinyang | 6.11 | 5.00% | 274 | 24.78 | 1 | -0.080 | -0.027 |
| 29 | Anqing | 5.31 | 1.57% | 83 | 23.61 | 1 | -0.022 | -0.012 |
| 30 | Jiujiang | 4.73 | 2.06% | 118 | 22.29 | 1 | -0.072 | -0.047 |
| / | Selected cities | 295.95 | 69.17% | 5790 | 3204.24 | 30 | -0.419 | / |
| Total | | | | | | $1+p = 31$ | $TE = -0.497$ | |

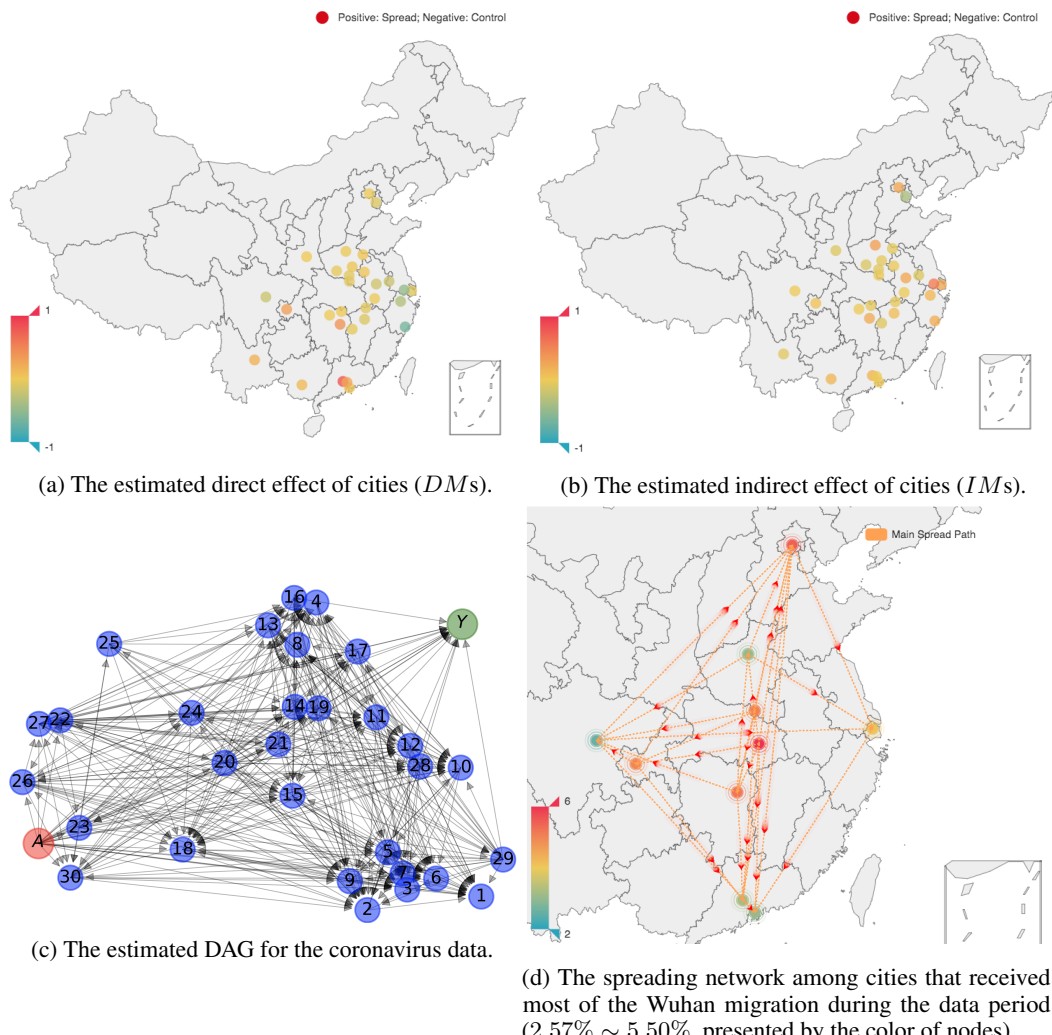

(a) The estimated direct effect of cities ($DM$s).

(b) The estimated indirect effect of cities ($IM$s).

(c) The estimated DAG for the coronavirus data.

(d) The spreading network among cities that received most of the Wuhan migration during the data period ($2.57\% \sim 5.50\%$, presented by the color of nodes).

Figure 18: Additional analysis of the causal effects of 2020 Hubei lockdowns on reducing the COVID-19 spread in China regulated by Chinese major cities outside Hubei.

also an almost all blue rectangle in the right top of the estimated matrix, showing non-central cities tend to have negative effects on central cities. Overall speaking, all the above finding accords with the migration trend during the Spring Festival period and intensive mutual communications among central cities in China, though there are also some noisy causal directions opposite with the main trend in each area due to possible confounders and identifiability issue in the linear Gaussian model.

In addition, we provide the estimated DAG for the coronavirus data in Figure 18c as the complete spreading network among major cities outside Hubei, where the exposure $A$, mediators $M$ (cities' corresponding index can be found in Table 6), and the outcome $Y$ are colored in red, blue and green, respectively. It can be observed that the in-degree is larger than the out-degree for nodes with small index, while an opposite rule is applied for nodes with large index. This finding is consistent with the migration trend identified in our main text. Lastly, we give the spreading network among cities that received most Wuhan migration during the data period, including Beijing, Shanghai, Guangzhou, Shenzhen, Chengdu, Chongqing, Zhengzhou, Changsha, and Xinyang, plus Wuhan, in Figure 18d, to illustrate the partial interaction trend among cities. Each node refers to a city with the color of the node presenting the percentage of received Wuhan migration, ranging from $2.57\%$ to $5.50\%$.

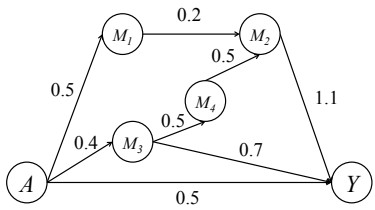

Figure 19: A weighted causal DAG $\mathcal{G}^{toy}$.

## E  TOY EXAMPLE

**Example E.1** *Here, we use a toy example of a weighted DAG $\mathcal{G}^{toy}$ under the LSEM given in Figure 19 to better demonstrate our definitions.*

*In $\mathcal{G}^{toy}$, $p = 4$ mediators are included, where $M_1 \leftarrow 0.2A + \epsilon_1$, $M_2 \leftarrow 0.2M_1 + 0.5M_4 + \epsilon_2$, $M_3 \leftarrow 0.4A + \epsilon_3$, $M_4 \leftarrow 0.5M_3 + \epsilon_4$, $Y = 0.5A + 1.1M_2 + 0.7M_3$. There are 4 directed path from $A$ to $Y$:*

*1) '$A \rightarrow Y$' with length 1;*
*2) '$A \rightarrow M_3 \rightarrow Y$' with length 2;*
*3) '$A \rightarrow M_1 \rightarrow M_2 \rightarrow Y$' with length 3;*
*4) '$A \rightarrow M_3 \rightarrow M_4 \rightarrow M_2 \rightarrow Y$' with length 4.*

*Since there exists 3 directed path $\pi^* \in \{\pi_{AY}(\mathcal{G}^{toy})\}$ such that the length of $\pi^*$ is larger than 2, we have the mediators in $\mathcal{G}^{toy}$ are interacted.*

*From the weighted DAG $\mathcal{G}^{toy}$, we have the direct effect of $A$ on $Y$ is $\gamma^{toy} = 0.5$, $\boldsymbol{\alpha}^{toy} \equiv [0.5, 0, 0.4, 0]^\top$, $\boldsymbol{\beta}^{toy} \equiv [0, 1.1, 0.7, 0]^\top$, and*

$$
B_M^{\top \, toy} = \begin{bmatrix} 0 & 0 & 0 & 0 \\ 0.2 & 0 & 0 & 0.5 \\ 0 & 0 & 0 & 0 \\ 0 & 0 & 0 & 0.5 \end{bmatrix}.
$$

*Note one may recover $\mathcal{G}^{toy}$ from the weight matrix $B^{toy}$ as long as the order of the vertices in $B^{toy}$ is given. Then, we have*

$$
(I_p - B_M^{\top \, toy})^{-1} = \begin{bmatrix} 1 & 0 & 0 & 0 \\ 0.2 & 1 & 0.25 & 0.5 \\ 0 & 0 & 1 & 0 \\ 0 & 0 & 0.5 & 1 \end{bmatrix}.
$$

*Thus, the indirect effect of $A$ on $Y$ is*

$$
\begin{aligned}
& \boldsymbol{\beta}^{toy \top}(I_p - B_M^{\top \, toy})^{-1}\boldsymbol{\alpha}^{toy} \\
=& [0, 1.1, 0.7, 0] \begin{bmatrix} 1 & 0 & 0 & 0 \\ 0.2 & 1 & 0.25 & 0.5 \\ 0 & 0 & 1 & 0 \\ 0 & 0 & 0.5 & 1 \end{bmatrix} [0.5, 0, 0.4, 0]^\top \\
=& [0, 1.1, 0.7, 0][0.5, 0.2, 0.4, 0.2]^\top \\
=& 0 + 0.22 + 0.28 + 0 = 0.5.
\end{aligned}
\tag{E.1}
$$

*From the Equation E.1 and Theorem 3.1, we have the DM of $M_2$ and $M_3$ is $0.22$ and $0.28$, respectively, while other DM are 0. Note that there is no explicit expression of the natural indirect effect through the directed path from $A$ to $Y$ (IM) due to the complex interaction among mediators, while we provide its theoretical form in Equation G.6 based on the path method in Wright (1921) and Nandy et al. (2017). Specifically, for each directed path from $A$ to $Y$, we have:*

*1) the direct effect through 'A → Y' (DE): 0.5;*

*2) the effect of path 'A → M_3 → Y': $0.4 \times 0.7 = 0.28$;*

*3) the effect of path 'A → M_1 → M_2 → Y': $0.5 \times 0.2 \times 1.1 = 0.11$;*

*4) the effect of path 'A → M_3 → M_4 → M_2 → Y': $0.4 \times 0.5 \times 0.5 \times 1.1 = 0.11$;*

*5) so the indirect effect of A on Y (IE) is $0.11 + 0.11 + 0.28 = 0.5$;*

*6) and the total effect of A on Y (TE) is $0.5 + 0.5 = 1.0$;*

*7) the indirect effect for $M_1$ ($IM_1$) corresponds to the effect of path 'A → M_1 → M_2 → Y', thus is 0.11;*

*8) the indirect effect for $M_2$ ($IM_2$) is zero since there is no path first goes through $M_2$ followed by other mediators;*

*9) the indirect effect for $M_3$ ($IM_3$) corresponds to the effect of path 'A → M_3 → M_4 → M_2 → Y', thus is 0.11;*

*10) the indirect effect for $M_4$ ($IM_4$) corresponds to the effect of path 'A → M_3 → M_4 → M_2 → Y', thus is 0.11.*

*We can calculate the last edge-specific effect directly from its definition. Since there is no $M_1 \to Y$ and $M_4 \to Y$ in $\mathcal{G}^{toy}$, we have $LE_1^{toy} = LE_4^{toy} = 0$. By deleting $M_2 \to Y$ in $\mathcal{G}^{toy}$, the total effect reduced by $0.11 + 0.11 = 0.22$, so $LE_2^{toy} = 0.22$; similarly, after deleting $M_3 \to Y$, the total effect decreases 0.28, thus $LE_3^{toy} = 0.28$. One may notice that the last edge-specific effects are equal to the $DMs$, and the additive of the last edge-specific effects is exact the last step of calculation $\boldsymbol{\beta}^{toy\top}(I_p - B_M^{\top\,toy})^{-1}\boldsymbol{\alpha}^{toy}$ in Equation E.1.*

## F  CONNECTION TO LITERATURE

We establish the connection between our proposed method to the literature from three different angles. First, we show that the individual mediation effect defined in Chakrabortty et al. (2018) can be decomposed into our defined $DM$ and $IM$ when the LSEM assumption holds. Next, we give an equivalent definition of the $DM$ through a type of special edge (last edge) in the causal graph. Lastly, we prove that the proposed $DM$ is consistent with the interventional effect via a particular mediator defined in Vansteelandt & Daniel (2017) under the LSEM.

### F.1  FROM INDIVIDUAL MEDIATION VIEWPOINT

Chakrabortty et al. (2018) defined the individual mediation effect under the LSEM as follows.

**Definition F.1**  *(Chakrabortty et al., 2018) Individual mediation effect for $M_i$:*

$$
\begin{aligned}
\eta_i = &\Big[E\{M_i|do(A = a + 1)\} - E\{M_i|do(A = a)\}\Big] \\
&\times \Big[E\{Y|do(M_i = m_i + 1)\} - E\{Y|do(M_i = m_i)\}\Big].
\end{aligned}
\tag{F.1}
$$

In the following theorem, we show that the summation of $\eta_i$ is strictly larger than the $IE$ if the mediators are not parallel. The proof is given in Section G.2.

**Theorem F.1** *If there exists at least one directed path $\pi^* \in \{\pi_{AY}(\mathcal{G})\}$ such that the length of $\pi^*$ is larger than 2, and the element in $B$ is nonnegative, then*

$$
\sum \eta_i > IE.
\tag{F.2}
$$

**Remark F.1** *From the above theorem, it is clear that the mediator effect defined in Chakrabortty et al. (2018) is not appropriate for interpreting the decomposition of the indirect effect, when there exists interaction among mediators (a common situation as described in the introduction). Here, we keep the condition that the element in $B$ is nonnegative, as the multiple count mediation effects in Chakrabortty et al. (2018) may cancel out in some cases and their summation would equal to the $IE$ by chance.*

Inspired by the proof of Theorem F.1, the mediator effect $\eta_i$ can be decomposed into two parts, the natural direct and indirect effect for $i$-th mediator, as shown in the following corollary.

**Corollary F.1** *Under assumptions (A1-A3) and Model 1, we have*

$$\eta_i = DM_i + IM_i. \tag{F.3}$$

**Remark F.2** *Corollary F.1 together with the definition of $\eta_i$ in Chakrabortty et al. (2018) provides a feasible way to numerically calculate the natural indirect effect $IM_i$. Specifically, by deleting the mediator $M_i$ in the causal graph, the reduced treatment effect corresponds to $\eta_i$, then $IM_i = \eta_i - DM_i$, where the explicit expression of the $DM_i$ is provided in Theorem 3.1. See more implementation details in Section B.*

## F.2 FROM GRAPHICAL PERSPECTIVE

Next, we give the definition of the edge-specific effect following Avin et al. (2005). Suppose a directed edge of interest as $X_i \to X_j$ in a weighted DAG $\mathcal{G}$. Define a new weighted DAG $\mathcal{G}'_{i,j}$ by deleting the directed edge $X_i \to X_j$ in $\mathcal{G}$, i.e. $\mathcal{G}'_{i,j} \equiv \mathcal{G} \setminus (X_i \to X_j)$.

**Definition F.2** *(Avin et al., 2005) Edge-specific effect:*

$$ET(X_i, X_j) = TE_{\mathcal{G}} - TE_{\mathcal{G}'_{i,j}}, \tag{F.4}$$

*where $TE_{\mathcal{G}}$ means the total effect in graph $\mathcal{G}$.*

We next give an equivalent definition of our proposed $DM$ from a graphical perspective. Let the edge in $\mathcal{G}$ that starts with $i$-th mediator and ends with node $Y$, i.e. $M_i \to Y$, as the $i$-th last edge. Denote the graph $\mathcal{G}$ deleting the $i$-th last edge ($M_i \to Y$) as $\mathcal{G}'_i$. We define the $i$th last edge-specific effect as

**Definition F.3** *Last edge-specific effect for $M_i$:*

$$LE_i = \begin{cases} TE_{\mathcal{G}} - TE_{\mathcal{G}'_i}, & \text{if there exists edge } M_i \to Y \text{ in } \mathcal{G}; \\ 0, & \text{otherwise.} \end{cases} \tag{F.5}$$

By Theorem 3.1, we have $(I_p - B_M^\top)^{-1}\boldsymbol{\alpha}$ is the causal effect of $A$ on $M$. Let $\zeta \equiv (I_p - B_M^\top)^{-1}\boldsymbol{\alpha}$, with its $i$-th element $\zeta_i \equiv \{(I_p - B_M^\top)^{-1}\boldsymbol{\alpha}\}_i$. Next, we show that the $i$-th last edge-specific effect can be presented as $\boldsymbol{\beta}_i\zeta_i$ under the LSEM in the following theorem, where $\boldsymbol{\beta}_i$ is the $i$-th element of the vector $\boldsymbol{\beta}$ and corresponds to the weight of the edge $M_i \to Y$. The proof can be found in Section G.3.

**Theorem F.2** *Under assumptions (A1-A3) and Model 1, we have*

$$LE_i = \boldsymbol{\beta}_i\zeta_i. \tag{F.6}$$

Based on Theorem 3.1, the natural direct effect of $M_i$ on $Y$ can be expressed as $DM_i = \boldsymbol{\beta}_i\zeta_i$. Thus, with the result of Theorem F.2, it is easy to show the following corollary.

**Corollary F.2** *Under assumptions (A1-A3) and Model 1, the natural direct effect of $M_i$ is equal to the $i$-th last edge-specific effect:*

$$LE_i = DM_i = \boldsymbol{\beta}_i\zeta_i. \tag{F.7}$$

**Remark F.3** *Here, both definitions describe the direct impact of one mediator $M_i$ on the outcome. The natural direct effect of a particular mediator $M_i$ can be understood as the influence when removing the direct edge between $M_i$ and $Y$. Thus, we have the equivalence between two definitions.*

Then, we can decompose the total natural indirect effect into $p$ last edge-specific effects or $p$ $DM$s as the following additive form, based on Theorem 3.2 and Corollary F.2.

**Corollary F.3** *Under assumptions (A1-A3) and Model 1, we have*

$$IE = \boldsymbol{\beta}^\top\zeta = \sum_{i=1}^{p}\boldsymbol{\beta}_i\zeta_i = \sum_{i=1}^{p}DM_i = \sum_{i=1}^{p}LE_i. \tag{F.8}$$

In fact, based on the uniqueness of each last edge, the natural indirect effect can be decomposed into $p$ last edge-specific effect regardless of the LSEM setting through the graphical perspective. We give the following intuitive conclusion. The proof can be found in Section G.4.

**Theorem F.3** *The $IE$ can be decomposed through $LEs$ as:*

$$IE = \sum_{i=1}^{p} LE_i. \tag{F.9}$$

**Remark F.4** *One can view the last edge-specific effect as the generalized definition of the natural direct effect for mediator without the LSEM assumption.*

### F.3 FROM INTERVENTIONAL EFFECT LEVEL

Finally, we show the consistency of our defined $DM$ to the interventional effect via a particular mediator defined in Vansteelandt & Daniel (2017) under the LSEM.

**Definition F.4** *(Vansteelandt & Daniel, 2017) Under assumptions (A1-A3), the interventional effect via $M_i$ is*

$$\xi_i = \sum_{m_1 \in \mathbb{M}_1} \cdots \sum_{m_p \in \mathbb{M}_p} \left[ E(Y|A = a, M_i = m_i, \Omega_i = o_i) P(\Omega_i = o_i | A = a) \right.$$
$$\left. \times \left\{ P(M_i = m_i | A = a+1) - P(M_i = m_i | A = a) \right\} \right], \tag{F.10}$$

*where $\mathbb{M}_i$ is the support of $M_i$, $o_i = [m_1, \cdots, m_{i-1}, m_{i+1}, \cdots, m_p]$, $P(M = m | A = a)$ is the probability of $M = m$ when setting $A = a$.*

**Theorem F.4** *Under assumptions (A1-A3) and Model 1, we have*

$$DM_i = \xi_i,$$

**Remark F.5** *The proof can be found in Section G.5. Based on Definition 3.2 and Equation F.10, both the proposed $DM$ and the effect defined in Vansteelandt & Daniel (2017) contain the information of the causal effect of $A$ on the mediator $M_i$, i.e. $P(M_i = m_i | A = a+1) - P(M_i = m_i | A = a)$.*

## G TECHNICAL PROOFS

### G.1 PROOF OF THEOREM 3.1

**Proof G.1** *In this proof, we will give the explicit expressions of causal effects defined under the LSEM. First, Equation 3 is equivalent to*

$$\begin{cases} A \equiv \epsilon_A, \\ M = \boldsymbol{\alpha}A + B_M^\top M + \epsilon_M, \\ Y = \gamma A + \boldsymbol{\beta}^\top M + \epsilon_Y. \end{cases} \tag{G.1}$$

*Based on $M = \boldsymbol{\alpha}A + B_M^\top M + \epsilon_M$, by moving $B_M^\top M$ to the left-hand side, we have*

$$(I_p - B_M^\top)M = \boldsymbol{\alpha}A + \epsilon_M.$$

*Suppose the mediators are sorted in the topological order (a series of elementary transformation of the matrix), then the matrix $B_M^\top$ is strictly upper triangular with the diagonal element as 0. Thus, we have $I_p - B_M^\top$ is invertible, then $I_p - B_M^\top$ under its original order should be also invertible (any invertible matrix after elementary transformation is still invertible).*

*Therefore, we can rewrite $M$ as a purely function of $A$ plus the error term as follows.*

$$M = (I_p - B_M^\top)^{-1} \boldsymbol{\alpha}A + (I_p - B_M^\top)^{-1} \epsilon_M. \tag{G.2}$$

*Then we replace mediators in Equation G.1 with Equation G.2 and obtain*

$$
\begin{cases}
A & \equiv \epsilon_A, \\
M & = (I_p - B_M^\top)^{-1}\boldsymbol{\alpha}A + (I_p - B_M^\top)^{-1}\epsilon_M, \\
Y & = \gamma A + \boldsymbol{\beta}^\top M + \epsilon_Y \\
& = \gamma A + \{\boldsymbol{\beta}^\top(I_p - B_M^\top)^{-1}\boldsymbol{\alpha}\}A + \{\boldsymbol{\beta}^\top(I_p - B_M^\top)^{-1}\epsilon_M + \epsilon_Y\}.
\end{cases}
\tag{G.3}
$$

*Next, we show how to get the explicit expressions of $E\{Y|do(A = a)\}$ under the LSEM. Following the results in Rosenbaum & Rubin (1983), under the assumption (A2), we have $P\{M|do(A = a)\} = P(M|A = a)$, and thus,*

$$
E\{M|do(A = a)\} = E(M|A = a).
$$

*Similarly, we can get $E\{Y|do(A = a)\} = E(Y|A = a)$ under the assumption (A1), and $E\{Y|do(A = a, M = m)\} = E(Y|A = a, M = m)$ under the assumption (A3).*

*Based on above results and Equation G.3, we have*

$$
\begin{aligned}
E\{Y|do(A = a)\} &= E\{Y|A = a\} \\
&= E\{\gamma A + \boldsymbol{\beta}^\top M + \epsilon_Y|A = a\} \\
&= \gamma a + \boldsymbol{\beta}^\top E\{M|A = a\} \\
&= \gamma a + \boldsymbol{\beta}^\top E\{(I_p - B_M^\top)^{-1}\boldsymbol{\alpha}A + (I_p - B_M^\top)^{-1}\epsilon_M|A = a\} \\
&= \gamma a + \boldsymbol{\beta}^\top(I_p - B_M^\top)^{-1}\boldsymbol{\alpha}a,
\end{aligned}
\tag{G.4}
$$

*where the first '=' is held under the assumption (A1), the second and forth '=' are given by Equation G.3 that $Y = \gamma A + \boldsymbol{\beta}^\top M + \epsilon_Y$ and $M = (I_p - B_M^\top)^{-1}\boldsymbol{\alpha}A + (I_p - B_M^\top)^{-1}\epsilon_M$.*

*Following the same calculation procedure of $E\{Y|do(A = a)\}$, we next give the natural direct effect under assumptions (A1-A3) and Model 1 as*

$$
\begin{aligned}
DE &= E\{Y|do(A = a + 1, M = m^{(a)})\} - E\{Y|do(A = a)\} \\
&= \{\gamma(a + 1) + \boldsymbol{\beta}^\top m^{(a)}\} - \{\gamma a + \boldsymbol{\beta}^\top m^{(a)}\} \\
&= \gamma,
\end{aligned}
$$

*where the first '=' is given by the definition of the DE.*

*Similarly, the natural indirect effect is*

$$
\begin{aligned}
IE &= E\{Y|do(A = a, M = m^{(a+1)})\} - E\{Y|do(A = a)\} \\
&= \{\gamma a + \boldsymbol{\beta}^\top m^{(a+1)}\} - \{\gamma a + \boldsymbol{\beta}^\top m^{(a)}\} \\
&= \boldsymbol{\beta}^\top(I_p - B_M^\top)^{-1}\boldsymbol{\alpha}(a + 1) - \boldsymbol{\beta}^\top(I_p - B_M^\top)^{-1}\boldsymbol{\alpha}a \\
&= \boldsymbol{\beta}^\top(I_p - B_M^\top)^{-1}\boldsymbol{\alpha}.
\end{aligned}
$$

*Thus, the total effect of $A$ on $Y$ is*

$$
TE = E\{Y|do(A = a + 1)\} - E\{Y|do(A = a)\} = DE + IE = \gamma + \boldsymbol{\beta}^\top(I_p - B_M^\top)^{-1}\boldsymbol{\alpha}.
$$

*Finally, we give the expression for the natural direct effect of $M_i$ on $Y$ under the LSEM. Based on the assumption (A2) and Equation G.2, we have*

$$
\begin{aligned}
&E\{M_i|do(A = a + 1)\} - E\{M_i|do(A = a)\} \\
&= E\{M_i|A = a + 1\} - E\{M_i|A = a\} \\
&= \{(I_p - B_M^\top)^{-1}\boldsymbol{\alpha}\}_i(a + 1) - \{(I_p - B_M^\top)^{-1}\boldsymbol{\alpha}\}_i a \\
&= \{(I_p - B_M^\top)^{-1}\boldsymbol{\alpha}\}_i,
\end{aligned}
\tag{G.5}
$$

*where $\{(I_p - B_M^\top)^{-1}\boldsymbol{\alpha}\}_i$ is the i-th element of the vector $(I_p - B_M^\top)^{-1}\boldsymbol{\alpha}$.*

*Then, based on $Y = \gamma A + \boldsymbol{\beta}^\top M + \epsilon_Y$ and the assumption (A3), we have,*

$$E\{Y|do(A = a, M_i = m_i^{(a)} + 1, \Omega_i = o_i^{(a)})\} - E\{Y|do(A = a)\}$$

$$=E\{Y|A = a, M_i = m_i^{(a)} + 1, \Omega_i = o_i^{(a)}\} - E\{Y|A = a\}$$

$$=\gamma a + \boldsymbol{\beta}^\top \begin{bmatrix} m_1^{(a)} \\ \vdots \\ m_i^{(a)} + 1 \\ \vdots \\ m_p^{(a)} \end{bmatrix} - \gamma a - \boldsymbol{\beta}^\top \begin{bmatrix} m_1^{(a)} \\ \vdots \\ m_i^{(a)} \\ \vdots \\ m_p^{(a)} \end{bmatrix} = \boldsymbol{\beta}^\top \boldsymbol{1}_i = \boldsymbol{\beta}_i,$$

*where $\boldsymbol{1}_i$ is a $p \times 1$ vector with the $i$-th element as 1 while others qual to 0, and $\boldsymbol{\beta}_i$ is the $i$-th element of the vector $\boldsymbol{\beta}$.*

*Thus, we have*

$$DM_i = \left[ E\{M_i|do(A = a + 1)\} - E\{M_i|do(A = a)\} \right]$$
$$\times \left[ E\{Y|do(A = a, M_i = m_i^{(a)} + 1, \Omega_i = o_i^{(a)})\} - E\{Y|do(A = a)\} \right],$$
$$= \{(I_p - B_M^\top)^{-1}\boldsymbol{\alpha}\}_i \times \boldsymbol{\beta}_i$$
$$= \boldsymbol{\beta}_i \{(I_p - B_M^\top)^{-1}\boldsymbol{\alpha}\}_i. \qquad \square$$

## G.2 PROOF OF THEOREM F.1

**Proof G.2** *1. If there is no directed path $\pi^* \in \{\pi_{AY}(\mathcal{G})\}$ such that the length of $\pi^*$ is larger than 2, i.e. the length of $\pi^* \in \{\pi_{AY}(\mathcal{G})\}$ is either 1 or 2. Here, the path with length 1 corresponds to $A \to Y$, and paths with length 2 are $A \to M_i \to Y$ with possibly $i = 1, \cdots, p$. Thus, there is no interaction among mediators.*

*By the definition of the LSEM, we have $B_M = \boldsymbol{0}_{p \times p}$, where $\boldsymbol{0}_{p \times p}$ is a $p \times p$ zero matrix. Following the path method (the causal effect of $X_i$ on $X_j$ along a directed path from $X_i \to X_j$ in $\mathcal{G}$ can be calculated by multiplying all edge weights along the path) illustrated in Wright (1921) and Nandy et al. (2017), we could obtain $\sum \eta_i = \sum_i \boldsymbol{\beta}_i \boldsymbol{\alpha}_i = IE$. (See a toy example provided in section E to illustrate how to use the path method to manually compute the causal effects.)*

*2. If there exists at least one directed path $\pi^* \in \{\pi_{AY}(\mathcal{G})\}$ such that the length of $\pi^*$ is larger than 2, and the element in B is nonnegative, we have $B_M \neq \boldsymbol{0}_{p \times p}$. Without loss of generality, suppose there exists $M_i \in M$ with a set of directed path that starts with A, contains $M_i$, then goes through other mediators, and ends with Y, denoted each path in such set as $\pi_{i,j} = \{A \to \cdots \to M_i \cdots \to \cdots \to Y\}$ for $j = 1, \cdots, n_i$, where $n_i$ is the size of such path set for $M_i$, and the weights of edges in $\pi_{i,j}$ is positive. Note the set $\{\pi_{i,j}\}$ excludes the paths end with $M_i \to Y$.*

*Let $e_{\pi_{i,j}}$ denote the causal effect of A on Y through directed path $\pi_{i,j}$. Based on the path method in Wright (1921) and Nandy et al. (2017) with the definition of $IM_i$, we have its theoretical form as*

$$IM_i = \sum_{j=1}^{n_i} e_{\pi_{i,j}}. \tag{G.6}$$

*By Equation G.5 and the definition of $\eta_i$, we have its first multiplier as*

$$E\{M_i|do(A = a + 1)\} - E\{M_i|do(A = a)\} = \{(I_p - B_M^\top)^{-1}\boldsymbol{\alpha}\}_i,$$

*which is also the first multiplier in both $DM_i$ and $IM_i$.*

*And the second multiplier of $\eta_i$ can be expressed as*

$$E\{Y|do(M_i = m_i + 1)\} - E\{Y|do(M_i = m_i)\}$$
$$=E\{Y|do(M_i = m_i^{(a)} + 1)\} - E\{Y|do(M_i = m_i^{(a)})\}$$
$$=E\{Y|do(A = a, M_i = m_i^{(a)} + 1)\} - E\{Y|do(A = a, M_i = m_i^{(a)})\},$$
$$=E\{Y|do(A = a, M_i = m_i^{(a)} + 1)\} - E\{Y|do(A = a)\},$$

where $m_i^{(a)}$ is the value of $M_i$ when setting $do(A = a)$. Here, the first '=' is valid since $m_i$ can be arbitrary number, and the second and third '=' are based on the equivalent interventions.

Based on the technique of plus and minus the same term, we decompose the second multiplier of $\eta_i$ into two parts as follows

$$
\begin{aligned}
&E\{Y|do(A = a, M_i = m_i^{(a)} + 1)\} - E\{Y|do(A = a)\} \\
&= \underbrace{\left[ E\{Y|do(A = a, M_i = m_i^{(a)} + 1, \Omega_i = o_i^{(a)})\} - E\{Y|do(A = a)\} \right]}_{\text{the second multiplier of } DM_i} \\
&+ \underbrace{\left[ E\{Y|do(A = a, M_i = m_i^{(a)} + 1)\} - E\{Y|do(A = a, M_i = m_i^{(a)} + 1, \Omega_i = o_i^{(a)})\} \right]}_{\text{the second multiplier of } IM_i}
\end{aligned}
$$

(G.7)

where $\Omega_i = M \setminus M_i$ is the sets of mediators except $M_i$, and $o_i^{(a)}$ is the value of $\Omega_i$ when setting $do(A = a)$. Here, the first term in the above equation corresponds to the second multiplier of $DM_i$, while the second term is the second multiplier of $IM_i$.

Thus, the summation of $\eta_i$ is

$$
\begin{aligned}
\sum \eta_i &= \sum_i \left\{ \left[ E\{M_i|do(A = a + 1)\} - E\{M_i|do(A = a)\} \right] \right. \\
&\qquad \left. \times \left[ E\{Y|do(M_i = m_i + 1)\} - E\{Y|do(M_i = m_i)\} \right] \right\} \\
&= \sum_i \{DM_i + IM_i\} = \sum_i DM_i + \sum_i IM_i = IE + \sum_i \sum_{j=1}^{n_i} e_{\pi_{i,j}},
\end{aligned}
$$

where the first '=' is from Definition F.1, the second '=' is given by Equation G.7 and Definition 3.2 and 3.3, and the last '=' comes from Theorem 3.2 and the theoretical form of $IM$ in Equation G.6.

Here, we have $e_{\pi_{i,j}} > 0$ since the weights of edges in $\pi_{i,j}$ is positive based on the path method in Wright (1921) and Nandy et al. (2017). Then, $\sum_i \sum_{j=1}^{n_i} e_{\pi_{i,j}}$ is also strictly larger than 0. Therefore, we have

$$
\sum \eta_i > IE. \qquad \square
$$

### G.3 PROOF OF THEOREM F.2

**Proof G.3** *1. If there doesn't exist edge $M_i \to Y$ in $\mathcal{G}$, then by definition we have $\beta_i = 0$. Thus, $LE_i = \beta_i \zeta_i = 0$.*

*2. If there exists edge $M_i \to Y$ in $\mathcal{G}$. Suppose there is a directed path set with size $m_i$ associated to the edge $M_i \to Y$, where each directed path $\tilde{\pi}_{i,j}$ starts with node $A$ and ends with $M_i \to Y$, denoted as $\tilde{\pi}_{i,j} = \{A \to \cdots \to \cdots \to M_i \to Y\}$ for $j = 1, \cdots, m_i$.*

*Let $e_{\tilde{\pi}_{i,j}}$ denote the causal effect of $A$ on $Y$ through directed path $\tilde{\pi}_{i,j}$, $e_{\tilde{\pi}_{i,j}}^{(A,M_i)}$ be the causal effect of $A$ on $M_i$ through directed path $\tilde{\pi}_{i,j}$, and $e^{(M_i,Y)}$ is the causal effect of $M_i$ on $Y$ through directed edge $M_i \to Y$. Following the path method in Wright (1921) and Nandy et al. (2017), we have $e_{\tilde{\pi}_{i,j}} = e_{\tilde{\pi}_{i,j}}^{(A,M_i)} e^{(M_i,Y)}$.*

*Then the $i$-th last edge-specific effect is equal to the summation of the effect through each path $\tilde{\pi}_{i,j}$, i.e.,*

$$
LE_i = \sum_{j=1}^{n_i} e_{\tilde{\pi}_{i,j}} = \sum_{j=1}^{n_i} e_{\tilde{\pi}_{i,j}}^{(A,M_i)} e^{(M_i,Y)} = e^{(M_i,Y)} \sum_{j=1}^{n_i} e_{\tilde{\pi}_{i,j}}^{(A,M_i)}.
$$

*Here, by the similar argument based on the path method, we have $e^{(M_i,Y)} = \beta_i$ and $\sum_{j=1}^{n_i} e_{\tilde{\pi}_{i,j}}^{(A,M_i)}$ as the total causal effect of $A$ on $M_i$.*

*Recall that $\zeta_i \equiv \{(I_p - B_M^\top)^{-1}\boldsymbol{\alpha}\}_i$ is the causal effect of $A$ on $M_i$. Therefore, the $i$-th LE is the product of the causal effect of $A$ on $M_i$ and the causal effect of $M_i$ on $Y$, i.e.,*

$$LE_i = \boldsymbol{\beta}_i \zeta_i. \qquad \square$$

## G.4 PROOF OF THEOREM F.3

**Proof G.4** *Given a general DAG $\mathcal{G}$ with nodes $\{A, M, Y\}$, let the union of all directed paths that contain the $i$-th last edge as $\tau_i = \{\pi : A \to \cdots \to M_i \to Y\}, i = 1, \cdots p$. Here, we have $\tau_i = \{\tilde{\pi}_{i,j}\}_{1 \leq j \leq m_j}$ established in Section G.3. It is clear that the union set of $\tau_i$ in $\mathcal{G}$ is equal to the set of all directed paths start with $A$ and end with node $Y$ (except $A \to Y$) in $\mathcal{G}$ as*

$$\bigcup_i \tau_i = \{\pi_{AY}(\mathcal{G})\} \setminus \{A \to Y\}.$$

*Also, based on the uniqueness of each last edge, $\tau_i$ is pairwise disjoint, i.e.*

$$\tau_i \bigcap \tau_j = \varnothing, \qquad \forall i \neq j.$$

*Since the $IE$ is defined as the total causal effect of $A$ on $Y$ that goes through mediators, we have the $IE$ equal to the causal effect that goes through the set $\{\pi_{AY}(\mathcal{G})\} \setminus \{A \to Y\}$, i.e. the $IE$ equal to the causal effect that goes through set $\bigcup_i \tau_i$. Based on the mutual disjoint property of $\tau_i$, we have the $IE$ is exactly the summation of the causal effect through $\tau_i$. Lastly, from the definition of $LE_i$, we have*

$$IE = \sum_{i=1}^p LE_i. \qquad \square$$

## G.5 PROOF OF THEOREM F.4

**Proof G.5** *The proof of the consistency of our defined $DM$ to the interventional effect $\xi_i$ can be completed based on Equation 3 under assumptions (A1-A3) and Model 1.*

*Recall the definition in Equation F.10, we have*

$$\xi_i = \sum_{m_1 \in \mathbb{M}_1} \cdots \sum_{m_p \in \mathbb{M}_p} \Bigg[ E(Y|A=a, M_i=m_i, \Omega_i=o_i)P(\Omega_i=o_i|A=a)$$

$$\times \Big\{ P(M_i=m_i|A=a+1) - P(M_i=m_i|A=a) \Big\} \Bigg].$$

$$= \sum_{m_1 \in \mathbb{M}_1} \cdots \sum_{m_p \in \mathbb{M}_p} \Big\{ E(Y|A=a, M_i=m_i, \Omega_i=o_i)P(\Omega_i=o_i|A=a)P(M_i=m_i|A=a+1)$$

$$- E(Y|A=a, M_i=m_i, \Omega_i=o_i)P(\Omega_i=o_i|A=a)P(M_i=m_i|A=a) \Big\}.$$

*Given $A = a$, the value of $M_i$ is $m_i^{(a)}$ and $\Omega_i$ takes $o_i^{(a)}$; while when setting $A = a + 1$, the value of $M_i$ is $m_{a+1}^{(i)}$. Therefore, we have $P(M_i = m_i|A = a) = 1$ if $m_i = m_i^{(a)}$ otherwise is 0, and $P(\Omega_i = o_i|A = a) = 1$ if $o_i = o_i^{(a)}$ otherwise is 0.*

*Under assumptions (A1-A3), we have*

$$\xi_i = E(Y|A=a, M_i=m_i^{(a+1)}, \Omega_i=o_i^{(a)}) - E(Y|A=a, M_i=m_i^{(a)}, \Omega_i=o_i^{(a)}).$$

*Then, based on the LSEM that $Y = \gamma A + \boldsymbol{\beta}^\top M + \epsilon_Y$, we can further obtain that*

$$\xi_i = \gamma a + \boldsymbol{\beta}^\top \begin{bmatrix} m_1^{(a)} \\ \vdots \\ m_i^{(a+1)} \\ \vdots \\ m_p^{(a)} \end{bmatrix} - \gamma a - \boldsymbol{\beta}^\top \begin{bmatrix} m_1^{(a)} \\ \vdots \\ m_i^{(a)} \\ \vdots \\ m_p^{(a)} \end{bmatrix} = \boldsymbol{\beta}_i \{m_i^{(a+1)} - m_i^{(a)}\}.$$

*From Equation G.2, we have*

$$\xi_i = \boldsymbol{\beta}_i \Big[ \{(I_p - B_M^\top)^{-1} \boldsymbol{\alpha}\}_i (a+1) - \{(I_p - B_M^\top)^{-1} \boldsymbol{\alpha}\}_i a \Big] = \boldsymbol{\beta}_i \{(I_p - B_M^\top)^{-1} \boldsymbol{\alpha}\}_i.$$

*Thus, under assumptions (A1-A3) and Model 1, we have*

$$DM_i = \xi_i. \qquad \square$$

