# OpenReview forum: "ANOCE: Analysis of Causal Effects with Multiple Mediators via Constrained Structural Learning"
_ICLR.cc/2021/Conference — ICLR 2021 Poster_

### Official Review · AnonReviewer1 · 2020-10-26
**ANOCE: Analysis of Causal Effects with Multiple Mediators via Constrained Structural Learning**

**Rating:** 6
**Confidence:** 2

**Review:**

### Summary
The authors propose a framework to analyse causal effects, considering interactions between mediators. The causal structure learning includes a temporal causal relationships of variables. Also, the authors validated the framework on real-world data set, in particular the Chinese coronavirus outbreak near Hubei.

### Reasons for score
The paper proposes an interesting and useful framework. The results show it can outperform the baselines (figure 3). The problem is clear and well motivated. Also, the technical details (section 4) are well explained, allowing for the reproducibility of the paper.

The proposed approach seems to be an extension of the DAG-GNN. Also, the latter is shown to have similar results in the analysis (fig 3). One main difference proposed by the authors is that ANOCE-CVAE can handle non-linear cases, contrarily to DAG-GNN. In [1] the authors mention that they method is actually able to handle parametric models beyond linear, so I may have missed something. Probably a way to improve the paper would be adding more details on the contribution with respect to the state of the art.

### Pros
- Analysis on synthetic and real data sets shows proposed model to outperform baselines.
- Detailed explanation on the technical part.

### Cons
- Not super clear what are the improvements with respect to the art.

### References
[1] Yu, Yue, et al. "Dag-gnn: Dag structure learning with graph neural networks." arXiv preprint arXiv:1904.10098 (2019).

---

### Official Review · AnonReviewer2 · 2020-10-28
**A valuable contribution for the field of causal inference**

**Rating:** 8
**Confidence:** 4

**Review:**

This paper presents a novel approach to structure learning for cause-effect models, focussing on the influence of mediators.

This approach is supported with an algorithm, which is implemented and tested on various datasets. The authors summarize important preliminaries and recap related approaches. The derivation is driven mathematically, supported by definitions and theorems.

It is remarkable that the authors build on Rubin's potential outcome framework and Pearl's graphical models and the do-formalism, where these "schools of thought" have been developing in somewhat seperated strains.

I recommend to accept the paper.

Overall, the paper is very well-structured, the accompanying appendix is detailed and provides extensive supplemental material.
The approach is driven theoretically and tested experimentally. \
The choice of COVID-19 data as experimental data adds a good final touch,  taking a recent and prevalent real-world example into account.

One downside is that no code implementation is provided (e.g. as supplementary material).

---

### Official Review · AnonReviewer4 · 2020-10-29
**A good paper with primarily incremental contributions**

**Rating:** 6
**Confidence:** 3

**Review:**

This paper combines two contributions to causal inference: a definition of and distinction between direct and indirect mediator effects to take account of cases of more than one mediator on a mediating path and how to identify them in linear structural equation models, and a modification of the DAG-GNN method for learning causal structure to incorporate certain background knowledge. In addition, the paper applies the concepts and the method to some interesting COVID19 data.

I am not sure how significant the first two contributions are; they strike me as primarily incremental. However, I am inclined to recommend acceptance in my overall judgment.

Some specific comments/questions:

1. Remark 3.2 is a bit misleading. Although the background setup is supposed to be a linear model, it's better to make the qualification clear *within the remark* that the decomposition refers to decomposition in linear models rather than decomposition in general.

2. The simulation uses primarily linear Gaussian models, whose structure is not always identifiable even given the background knowledge assumed in this paper. That is, sometimes the direction of an edge between two mediators may not be identifiable from observational data. It's unclear to me how this issue is handled in the experiments. An arbitrary member in the equivalence class is used to estimate the various effects?

3. Similarly, with the COVID19 data, the determination of some of the directions in the discovered network may be quite arbitrary. Did the authors notice any causal direction that does not make much sense or accord with the known migration trend?

4. The method is to said to also apply when the exposure is categorical. I would appreciate some clarification on how this variation affects the definitions and the identification formulas. And how about categorical mediators?

5. On p. 3 there is the following remark on methods based on conditional independence tests: "However, these methods have no guarantee in the finite sample regime because of the hardness of testing conditional independence when the variables are continuous (Shah & Peters, 2018)." This remark is a little misleading. The lack of finite sample guarantee in the absence of substantial assumptions is not just a problem with such methods. It is everyone's problem, as argued in, for example, Robins et al. (2003) "Uniform consistency in causal inference".

---

### Official Review · AnonReviewer3 · 2020-10-30
**ICLR 2021 Review of paper 1394: ANOCE: Analysis of Causal Effects with Multiple Mediators via Constrained Structural Learning**

**Rating:** 5
**Confidence:** 4

**Review:**

##### 1. Summary
The paper proposes a framework for the analysis of causal inference. Its main contribution is to decompose the indirect effect by teasing out the causal contribution of a set of mediators. In a series of experiments with simulated data the authors show that the proposed method, ANOCE, outperforms other comparison partners. The manuscript also contains an analysis of real-world data that describes the causal effects of the lockdown of cities in the Hubei province (China) to reduce the spread of COVID-19. An extensive supplementary file is also part of the submission and it includes additional experiments and technical proofs.

##### 2. Rationale for the score
The paper is well structured and very well written. The main document provides all explicit definitions of the causal effects under the linear structural equation model (LSEM). The appendix contains additional technical proofs. I find the notation to be consistent and easy to follow. Despite these pros, I find the results to be inconclusive and I suggest additional experiments that will make a more persuasive case for the method.

##### 3. Positive aspects
- the problem is well formulated and the manuscript is clear and well written
- the authors show how factors that affect the performance of other methods, such as the number of mediators or sparisity, have no negative impact on the performance of ANOCE.
- Tables 2 and 3 in the appendix provide a convincing argument on how ANOCE can disentangle the natural direct effect of the mediators as the sample size increases.
- I commend the authors for using ANOCE to analyze real-world data. The analysis of COVID-19 spread in China is interesting, although some aspects of the results do not seem convincing (see section 4 below for more details).

##### 4. Negative aspects
- For the comparison partners, the text indicates that default hyper-parameters were used. It is not clear for which methods the parameters were fine tuned, if this was done at all. As shown in Table 4, NOTEARS and DAG-GNN perform worse than PC in some settings of scenario 3*, with respect to structural Hamming distance. This begs the following questions: i) for the VAE in DAG-GNN, what are the default parameters? ii) for NOTEARS, Zheng et al. performed an analysis of the sensitivity of the threshold. They report a choice of w = 0.3 for their experiments, making clear that the choice of threshold becomes problematic as the sample size shrinks. Although the authors of the current manuscript also chose 0.3 as threshold, the question is how appropriate is this threshold for the simulated dataset they use. Note: In contrast to the 500 samples generated in the current manuscript, 1,000 simulated samples were created in Zheng et al. 2018. A fair comparison would have been to compare against the same simulated datasets described in Zheng et al. 2018 (see details below).
- In my opinion, some additional experiments are needed to create a convincing case for ANOCE. From the description in the Appendix (Section C.1) it seems that only one random graph was generated. Of course, it is understandable if the plots shown in Figures 2 and 3 only depict one true graph. But in order to compare against other methods, the experiments must consist of hundreds of different random graphs. Two considerations: i) the expected number of edges for each node is set to 2. Why not try other settings where the expectation is 1, or 4? ii) the random graphs are created using the Erdös-Rényi model. To make a stronger case for the method, the experiments should contain random graphs generated as scale-free networks, as detailed in [R1]. Combining i) and ii), these random networks are usually named ER1, ER2, ER4, SF1, SF2, and SF4.
- With respect to the analysis of COVID-19 spread, there are some aspects that are not clear: i) how to interpret the values in Table 5. For example, Beijing and Changsha have a similar value in the “Ratio of Wuhan” column. The IM of both cities is also similar but the DM of Beijing is close to null. It is not clear if the DM and IM values can be interpreted properly. Therefore, this makes me question the added value of this table. ii) Figure 10.c is too small to see and no meaningful insights can be extracted from it. One suggestion: if each node is linked to a real city, plot the figure on a map of China. The nodes A and Y can be outside the map (similar to Figure 10.d). iii) what do the colors mean in Figure 10.d?

##### 5. Questions to be addressed during rebuttal period
Unless I’m mistaken, the current results seem to be for only one random ER2 network (one with 50 nodes and another with 500 nodes, to be precise). Please, clarify the generation process for random network(s).

Additionally, it will be useful to learn how ANOCE performs in other types of networks (ER1, ER4, SF1, SF2 and SF4).

What hyper-parameters were used for the comparison partners?

##### 6. Additional references
[R1] Barabási, Albert-László, and Réka Albert. "Emergence of scaling in random networks." Science 286, no. 5439 (1999): 509-512.

---

### Decision · Program_Chairs · 2021-01-07
**Final Decision**

**Decision:**

Accept (Poster)

**Comment:**

This paper proposes an approach to learn the causal structure underlying a dataset with acyclicity and other structure constraints, and then used the inferred structure to compute partial causal effects. The authors show that, on simulated data, the proposed method outperforms others in the literature. The manuscript also contains an analysis of real-world data that describes the causal effects of the lockdown of cities in the Hubei province (China) to reduce the spread of COVID-19.

Overall, the reviewers think that this is a well structured and written paper. From a novelty viewpoint, the main contribution consists in formalising the causal contribution of mediators, as the method for computing the causal structure is based on a small modification to previous literature.

The main concern raised by the reviewers were on the experimental evaluation. Some of these concerns were addressed by the authors during rebuttal, whilst some on the number of nodes remained. We encourage the authors to consider these concerns in the final version of the manuscript.